# $z$-SIGNFEDAVG: A UNIFIED STOCHASTIC SIGN-BASED COMPRESSION FOR FEDERATED LEARNING

## ABSTRACT

Federated Learning (FL) is a promising privacy-preserving distributed learning paradigm but suffers from high communication cost when training large-scale machine learning models. Sign-based methods, such as SignSGD (Bernstein et al., 2018), have been proposed as a biased gradient compression technique for reducing the communication cost. However, sign-based algorithms could diverge under heterogeneous data, which thus motivated the development of advanced techniques, such as the error-feedback method and stochastic sign-based compression, to fix this issue. Nevertheless, these methods still suffer from slower convergence rates. Besides, none of them allows multiple local SGD updates like FedAvg (McMahan et al., 2017). In this paper, we propose a novel noisy perturbation scheme with a general symmetric noise distribution for sign-based compression, which not only allows one to flexibly control the tradeoff between gradient bias and convergence performance, but also provides a unified viewpoint to existing stochastic sign-based methods. More importantly, the unified noisy perturbation scheme enables the development of the very first sign-based FedAvg algorithm ($z$-SignFedAvg) to accelerate the convergence. Theoretically, we show that $z$-SignFedAvg achieves a faster convergence rate than existing sign-based methods and, under the uniformly distributed noise, can enjoy the same convergence rate as its uncompressed counterpart. Extensive experiments are conducted to demonstrate that the $z$-SignFedAvg can achieve competitive empirical performance on real datasets and outperforms existing schemes.

## 1 INTRODUCTION

We consider the Federated Learning (FL) network with one parameter server and $n$ clients (McMahan et al., 2017; Li et al., 2020a), with the focus on solving the following distributed learning problem

$$\min_{x \in \mathbb{R}^d} f(x) = \frac{1}{n} \sum_{i=1}^{n} f_i(x), \tag{1}$$

where $f_i(\cdot)$ is the local objective function for the $i$-th client, for $i = 1, \ldots, n$. Throughout this paper, we assume that each $f_i$ is smooth and possibly non-convex. The local objective functions are generated from the local dataset owned by each client. When designing distributed algorithms to solve (1), a crucial aspect is the communication efficiency since a massive number of clients need to transmit their local gradients to the server frequently (Li et al., 2020a). As one of the most popular FL algorithms, the federated averaging (FedAvg) algorithm (McMahan et al., 2017; Konečný et al., 2016) considers multiple local SGD updates with periodic communications to reduce the communication cost. Another way is to compress the local gradients before sending them to the server (Li et al., 2020a; Alistarh et al., 2017; Reisizadeh et al., 2020). Among the existing compression methods, a simple yet elegant technique is to take the sign of each coordinate of the local gradients, which requires only one bit for transmitting each coordinate. For any $x \in \mathbb{R}$, we define the sign operator as: $\text{Sign}(x) = 1$ if $x \geq 0$ and $-1$ otherwise.

It has been shown recently that optimization algorithms with the sign-based compression can enjoy a great communication efficiency while still achieving comparable empirical performance as uncompressed algorithms (Bernstein et al., 2018; Karimireddy et al., 2019; Safaryan & Richtárik, 2021). However, for distributed learning, especially the scenarios with heterogeneous data, i.e.,

$f_i \neq f_j$ for every $i \neq j$, a naive application of the sign-based algorithm may end up with divergence (Karimireddy et al., 2019; Chen et al., 2020a; Safaryan & Richtárik, 2021).

**A counterexample for sign-based distributed gradient descent.** Consider the one-dimensional problem with two clients: $\min_{x \in \mathbb{R}} (x - A)^2 + (x + A)^2$, where $A > 0$ is some constant. For any $x \in [-A, A]$, the averaged sign gradient at $x$ is $\text{Sign}(x - A) + \text{Sign}(x + A) = 0$, i.e., the algorithm never moves. Similar examples are also discussed by (Chen et al., 2020a; Safaryan & Richtárik, 2021). The fundamental reason for this undesirable result is the uncontrollable bias brought by the sign-based compression.

There are mainly two approaches to fixing this issue in the existing literature. The first one is the stochastic sign-based method, which introduces stochasticity into the sign operation (Jin et al., 2020; Safaryan & Richtárik, 2021; Chen et al., 2020a), and the second one is the Error-Feedback (EF) method (Karimireddy et al., 2019; Vogels et al., 2019; Tang et al., 2019). However, these works are still unsatisfactory. Specifically, on one hand, both the theoretical convergence rates and empirical performance of these algorithms are still worse than uncompressed algorithms like (Ghadimi & Lan, 2013; Yu et al., 2019). On the other hand, none of them allows the clients to have multiple local SGD updates within one communication round like the FedAvg, which thereby are less communication efficient. This work aims at addressing these issues and closing the gaps for sign-based methods.

**Main contributions.** Our contributions are summarized as follows.

(1) **A unified family of stochastic sign operators.** We show an intriguing fact: The bias brought by the sign-based compression can be flexibly controlled by injecting a proper amount of random noise before the sign operation. In particular, our analysis is based on a novel noisy perturbation scheme with a general symmetric noise distribution, which also provides a unified framework to understand existing stochastic sign-based methods including (Jin et al., 2020; Safaryan & Richtárik, 2021; Chen et al., 2020a).

(2) **The first sign-based FedAvg algorithm.** In contrast to the existing sign-based methods which do not allow multiple local SGD updates within one communication round, based on the proposed stochastic sign-based compression, we design a novel family of sign-based federated averaging algorithms ($z$-SignFedAvg) that can achieve the best of both worlds: high communication efficiency and fast convergence rate.

(3) **New theoretical convergence rate analyses.** By leveraging the asymptotic unbiasedness property of the stochastic sign-based compression, we derive a series of theoretical results for $z$-SignFedAvg and demonstrate its improved convergence rates over the existing sign-based methods. In particular, we show that by injecting a sufficiently large uniform noise, $z$-SignFedAvg can have a matching convergence rate with the uncompressed algorithms.

**Organization.** In Section 2, the proposed general noisy perturbation scheme for the sign-based compression and its key property, i.e., asymptotic unbiasedness, are presented. Inspired by this result, the main algorithms are devised in Section 3 together with their convergence analyses under different noise distribution parameters. We evaluate our proposed algorithms on real datasets and benchmarks with existing sign-based methods in Section 4. Finally, conclusions are drawn in Section 5.

**Notations.** For any $x \in \mathbb{R}^d$, we denote $x(j)$ as the $j$-th element of the vector $x$. We define the $\ell_p$-norm for $p \geq 1$ as $\|x\|_p = (\sum_{j=1}^{d} |x(j)|^p)^{\frac{1}{p}}$. We denote that $\|\cdot\| = \|\cdot\|_2$, and $\|x\|_\infty = \max_{j \in \{1,...,d\}} |x(j)|$. For any function $f(x)$, we denote $f^{(k)}(x)$ as its $k$-th derivative, and for a vector $x = [x(1), ..., x(d)]^\top \in \mathbb{R}^d$, we define $\text{Sign}(x) = [\text{Sign}(x(1)), ..., \text{Sign}(x(d))]^\top$.

## 1.1 RELATED WORKS

**Stochastic sign-based method.** Our proposed algorithm belongs to this category. Among the existing works (Safaryan & Richtárik, 2021; Jin et al., 2020; Chen et al., 2020a), the setting considered by (Safaryan & Richtárik, 2021) is closest to ours since the latter two consider gradient compression not only in the uplink but also in the downlink. Despite of this difference and the use of different convergence metrics, the algorithms therein achieve the same convergence rate $O(\tau^{-\frac{1}{4}})$, where $\tau$ is the total number of gradient queries to the local objective function. Compared to existing works, our proposed $z$-SignFedAvg requires a slightly stronger assumption on the mini-batch gradient noise, but achieves a faster convergence rate $O(\tau^{-\frac{1}{3}})$ or even $O(\tau^{-\frac{1}{2}})$, with the standard squared $\ell_2$-norm of gradients as the convergence metric.

**Error-Feedback method.** The error-feedback (EF) method is first proposed by (Seide et al., 2014) and later theoretically justified by (Karimireddy et al., 2019). Then, (Vogels et al., 2019; Tang et al., 2019; 2021a) further extended this EF method into distributed and adaptive gradient schemes. The key idea of the EF-based methods is to show that the sign operator scaled by the gradient norm is a contractive compressor, and the error induced by the contractive compressor can be compensated. However, such EF-based methods cannot deal with partial client participation otherwise the error residuals cannot be correctly tracked. Besides, the EF-based methods have a convergence rate $\mathcal{O}(\tau^{-\frac{1}{2}} + d^2\tau^{-1})$, where $d$ is the dimension of the gradients, and therefore is not competitive for high-dimension problems.

**Unbiased quantization method.** Apart from the sign-based gradient compression, another popular way of compression is the unbiased stochastic quantization method adopted by (Alistarh et al., 2017; Reisizadeh et al., 2020; Haddadpour et al., 2021). A key assumption made by this category of methods is that the quantization error is bounded by the norm of the input, which however does not hold for sign-based compression, and therefore the existing convergence results therein do not apply to sign-based methods. Besides, as shown in (Alistarh et al., 2017; Reisizadeh et al., 2020), these methods usually have degraded convergence speed when fewer quantization bits are used.

As mentioned, some of the existing sign-based methods like (Chen et al., 2020a; Safaryan & Richtárik, 2021) do not adopt the standard squared $\ell_2$-norm of gradients as the metric for the convergence rate analysis. Thus, it is tricky to make a fair comparison between them and the proposed $z$-SignFedAvg. In Appendix A, we provide a detailed discussion and summarize the convergence rates of some representative algorithms in Table 2.

## 2 SIGN OPERATOR WITH SYMMETRIC AND ZERO-MEAN NOISE

In this section, we introduce a general noisy perturbation scheme for the sign-based compression and analyze the asymptotic unbiasedness of compressed gradients. The results serve as the foundation for the proposed algorithms in subsequent sections.

**Key observation.** Let $\xi$ be a random variable that is symmetric, zero-mean and has the p.d.f $p(t)$. If $p(0) \neq 0$ and $p(t)$ is continuous and uniformly bounded on $(-\infty, +\infty)$, then it holds that

$$\lim_{\sigma \to +\infty} \frac{\sigma}{2p(0)} \mathbb{E}[\text{Sign}(x + \sigma\xi)] = \lim_{\sigma \to +\infty} \frac{\sigma}{p(0)} \int_0^{\frac{x}{\sigma}} p(t)dt = x. \tag{2}$$

In other words, the perturbed sign operator is an asymptotically unbiased estimator of the input $x$ when $\sigma \to \infty$. Furthermore, assume that $p(t)$ is uniformly bounded on $(-\infty, +\infty)$ and differentiable for an arbitrary order. Then, with the Taylor's expansion, we can have $\frac{\sigma}{p(0)} \int_0^{\frac{x}{\sigma}} p(t)dt = x + \frac{1}{p(0)} \sum_{k=1}^{+\infty} \frac{p^{(k)}(0)x^{k+1}}{(k+1)!\sigma^k} = x + \sum_{k=1}^{+\infty} p^{(k)}(0)\mathcal{O}\left(\sigma^{-k}\right)$. Therefore, suppose that $K$ is the largest integer such that $p^{(1)}(0) = 0, ..., p^{(K)}(0) = 0$. The LHS of (2) will converge to $x$ with the order $\mathcal{O}(\sigma^{-(K+1)})$. This observation motivates us to propose the following family of noise distribution parameterized by a positive integer $z \in \mathbb{Z}_+$.

**Definition 1** ($z$-distribution). *A random variable $\xi_z$ is said to follow the $z$-distribution if its p.d.f is*

$$p_z(t) = \frac{1}{2\eta_z} e^{-\frac{t^{2z}}{2}}, \tag{3}$$

*where $\eta_z = 2^{\frac{1}{2z}} \Gamma\left(1 + \frac{1}{2z}\right)$ and $\Gamma(z) = \int_0^{+\infty} t^{z-1}e^{-t}dt$ is the Gamma function.*

It can be verified $p_z(t)$ in (3) is a valid p.d.f. When $z = 1$, it corresponds to the standard Gaussian distribution. In addition, one can also show that $p_z(t)$ converges to the p.d.f of the uniform random variable on the interval $[-1, 1]$ when $z \to +\infty$ (see Lemma 2 in Appendix B). This $z$-distribution has a nice property that can be leveraged to bound the bias caused by the sign-based compression, as stated in the following lemma.

**Lemma 1.** *For any $x \in \mathbb{R}^d$ and $\sigma > 0$,*

$$\left\| \eta_z \sigma \mathbb{E}\left[\text{Sign}(\mathbf{x} + \sigma\xi_\mathbf{z})\right] - x \right\|^2 \leq \frac{\|x\|_{4z+2}^{4z+2}}{4(2z+1)^2\sigma^{4z}}, \tag{4}$$

*where $\xi_z(1), ..., \xi_z(d)$ follow the i.i.d. $z$-distribution.*

**Remark 1.** *One can see that the RHS of* (4) *involves the term* $(\|x\|_{4z+2}/\sigma)^{4z}$. *Thus, as long as* $\sigma > \|x\|_\infty$, *the LHS of* (4) *converges to zero when* $z \to +\infty$. *Since Lemma 2 implies that* $\xi_\infty$ *follows the i.i.d uniform distribution on* $[-1, 1]$, *we obtain* $\sigma \mathbb{E}\left[\text{Sign}(x + \sigma \xi_\infty)\right] = x$ *as long as* $\sigma > \|x\|_\infty$. *It is interesting to remark that the stochastic sign operators proposed in (Jin et al., 2020; Safaryan & Richtárik, 2021) are exactly the sign operator injected by the uniform noise, and (Chen et al., 2020a) also considered the use of a symmetric noise for gradient perturbation. Thus, sign-based compression with the* $z$-*distribution offers a unified perspective to understand the relationship among the existing stochastic sign-based methods.*

## 3  $z$-SIGNFEDAVG ALGORITHM

In this section, based on the analysis in Section 2, we propose the following sign-based FedAvg algorithm, termed as $z$-SignFedAvg. While FedAvg-type algorithms with gradient compression are also presented in (Haddadpour et al., 2021), they require unbiased compression and are not applicable to sign-based methods. The details of $z$-SignFedAvg are presented in Algorithm 1. A prominent difference between the proposed $z$-SignFedAvg and the existing sign-based methods lies in that the clients are allowed to perform multiple SGD updates per communication round ($E > 1$) before applying the stochastic sign-based compression. Like the FedAvg algorithm, it is anticipated that $z$-SignFedAvg can greatly benefit from this and has a significantly reduced communication cost.

Note that in practice we only consider $z = 1$ and $z = +\infty$ for the $z$-SignFedAvg since they correspond to the Gaussian distribution and uniform distribution, respectively. Nevertheless, we are interested in the convergence properties of $z$-SignFedAvg for a general positive integer $z$ as it provides better insights on the role of $z$ for the convergence rate.

---

**Algorithm 1** $z$-SignFedAvg (or $z$-SignSGD when $E = 1$)

**Require:** Total communication rounds $T$, number of local steps $E$, number of clients $n$, clients stepsize $\gamma$, server stepsize $\eta$, noise coefficient $\sigma$, parameter of noise distribution $z$.
1: Initialize $x_0$.
2: **for** $t = 1$ to $T$ **do**
3:     **On Clients:**
4:     **for** $i = 1$ to $n$ **do**
5:         $x_{t-1,0}^i = x_{t-1}$
6:         **for** $s = 1$ to $E$ **do**
7:             $g_{t-1,s}^i = g_i(x_{t-1,s-1}^i)$, where $g_i(\cdot)$ is the mini-batch gradient oracle of the $i$-th client.
8:             $x_{t-1,s}^i = x_{t-1,s-1}^i - \gamma g_{t-1,s}^i$.
9:         **end for**
10:         $\Delta_{t-1}^i = \text{Sign}\left(\frac{x_{t-1} - x_{t-1,E}^i}{\gamma} + \sigma \xi_z\right)$, where $\xi_z(1), ..., \xi_z(d) \sim p_z(t)$ i.i.d.
11:         Send $\Delta_{t-1}^i$ to the server.
12:     **end for**
13:     **On Server:**
14:     $x_t = x_{t-1} - \eta \gamma \frac{1}{n} \sum_{i=1}^n \Delta_{t-1}^i$.
15:     Broadcast $x_t$ to the clients.
16: **end for**
17: **return** $x_T$.

---

We first state some standard assumptions for problem (1).

**Assumption 1.** *We assume that each* $f_i(x)$ *has the following properties:*

> *A.1 We can access a mini-batch gradient oracle that is unbiased and has bounded variance, i.e.,* $\mathbb{E}[g_i(x)] = \nabla f_i(x)$ *and* $\mathbb{E}[\|g_i(x) - \nabla f_i(x)\|_2^2] \leq \zeta^2$.

> *A.2 Each* $f_i$ *is smooth, i.e., for any* $x, y \in \mathbb{R}^d$, *there exists some non-negative constants* $L_1, \ldots, L_d$, *such that* $f(y) - f(x) \leq \langle \nabla f(x), y - x \rangle + \frac{\sum_{j=1}^d L_j(y(j)-x(j))^2}{2}$.

> *A.3 There exists some constant* $f^*$ *such that* $f(x) \geq f^*, \forall x \in \mathbb{R}^d$.

> *A.4 There exists a constant* $G \geq 0$ *such that* $\|\nabla f_i(x)\| \leq G, \forall i = 1, ..., n,$ *and* $x \in \mathbb{R}^d$.

Assumption A.2 is a more fine-grained assumption on the function smoothness than the commonly used one and is also used by (Bernstein et al., 2018; Safaryan & Richtárik, 2021). For the convergence rate analysis, we consider two cases, namely, the case with $z < +\infty$ and the case of $z = \infty$.

### 3.1 CASE 1: $z < +\infty$

As we can see from Lemma 1, there always exists some gradient bias when $z < +\infty$. In order to bound it, we further assume that a higher order moment of the mini-batch gradient noise is bounded.

**Assumption 2.** *There exists a constant $Q_z \geq 0$ such that for any $x \in \mathbb{R}^d$, we have*

$$\mathbb{E}[\|g_i(x) - \nabla f_i(x)\|_{4z+2}^{4z+2}] \leq Q_z. \tag{5}$$

**Theorem 1.** *Suppose that Assumption 1 and 2 hold. Denote $\bar{x}_{t,s} = \frac{1}{n}\sum_{i=1}^n x_{t,s}^i$ and $L_{\max} = \max_j L_j$. Then, for $\eta = \eta_z \sigma$, $\gamma \leq \frac{1}{L_{\max}}$ and $z < +\infty$ in Algorithm 1, we have*

$$\mathbb{E}\left[\frac{1}{TE}\sum_{t=1}^T\sum_{s=1}^E \|\nabla f(\bar{x}_{t-1,s-1})\|^2\right] \leq \underbrace{\frac{2\mathbb{E}[f(x_0) - f^*]}{TE\gamma} + \frac{\gamma\zeta^2 L_{\max}}{n} + \frac{4\gamma^2(E-1)EL_{\max}^2(\zeta^2 + G^2)}{3}}_{\text{(a) Standard terms in FedAvg}}$$

$$\tag{6a}$$

$$+ \underbrace{\frac{2^{2z+1}E^{2z}\sqrt{Q_z + G^{4z+2}}G}{\sqrt{2}(2z+1)\sigma^{2z}} + \frac{\gamma 2^{4z}E^{4z+1}(Q_z + G^{4z+2})L_{\max}}{2(2z+1)^2\sigma^{4z}}}_{\text{(b) Bias terms}}$$

$$\tag{6b}$$

$$+ \underbrace{\frac{4\eta_z^2\gamma\sigma^2\sum_{j=1}^d L_j}{En}}_{\text{(c) Variance term}}. \tag{6c}$$

**When is the bound non-trivial?** Since we assume that the $\ell_2$-norm of gradient is bounded by $G$, all the terms in the RHS of (6) should be no larger than $G^2$. For example, to have the first term in (6b) less than $G^2$, one requires $\sigma$ to be greater than $2^{1+\frac{1}{4z}}E\left(Q_z/G + G^{4z}\right)^{\frac{1}{4z}}/(2z+1)^{\frac{1}{2z}}$.

**Bias-variance trade-off.** An interesting observation from Theorem 1 is that there exists a trade-off between the bias and variance terms. One can see that the terms in (6b) is caused by the gradient bias of the sign operation (see (4)) and is an infinitesimal of $\sigma$ with $\mathcal{O}\left(\sigma^{-2z}\right)$, while the term in (6c) is due to the injected noise and is in the order of $\mathcal{O}\left(\gamma\sigma^2\right)$. Specifically, the first term in (6b) only depends on the noise scale $\sigma$ and mostly affects the final objective. Meanwhile, the variance term in (6c) mainly affects the convergence speed because a smaller stepsize is required for it to diminish.

Theoretically, we can choose an iteration-dependent noise scale $\sigma$ so as to make the algorithm converge to a stationary solution. To see this, let us denote $\tau = TE$ as the total number of gradient queries per client, and present the following corollary.

**Corollary 1** (Informal). *Let $\gamma = \min\{n^{\frac{z}{2z+1}}\tau^{-\frac{z+1}{2z+1}}, L_{\max}^{-1}\}$ and $\sigma = (n\tau)^{\frac{1}{4z+2}}$ in Theorem 1, and let $E \leq n^{-\frac{3z}{4z+2}}\tau^{\frac{z+2}{4z+2}}$. We have*

$$\mathbb{E}\left[\frac{1}{\tau}\sum_{t=1}^T\sum_{s=1}^E \|\nabla f(\bar{x}_{t-1,s-1})\|^2\right] = \mathcal{O}\left((n\tau)^{-\frac{z}{2z+1}}\right). \tag{7}$$

**Achieveing linear speedup.** From Corollary 1, we can see that the $z$-SignFedAvg needs $(n\tau)^{\frac{3z}{4z+2}}$ communication rounds to achieve a linear-speedup convergence rate. Particularly, when $z = 1$, the corresponding convergence rate is $\mathcal{O}((n\tau)^{-\frac{1}{3}})$ and the required communication rounds is $(n\tau)^{\frac{1}{2}}$. To the best of our knowledge, the previous works have never shown the sign-based method can achieve a linear-speedup convergence rate.

**Relationship to (Chen et al., 2020a).** The work (Chen et al., 2020a) also considered the use of a symmetric and zero-mean noise for the sign-based compression and proved that the algorithm has a convergence rate $\mathcal{O}(\tau^{-\frac{1}{4}})$. However, their results have three differences from our $z$-SignFedAvg and Theorem 1. First, (Chen et al., 2020a) considered gradient compression both in the uplink and downlink communications. In addition, the convergence metric they used is not the standard squared $\ell_2$-norm of gradients and is hard to interpret. Second, their analysis is rooted in the median-based algorithm, whereas we judiciously exploit the property of the sign operation and hence provide a general analysis framework for the stochastic sign-based methods. Last but not the least, unlike our $z$-SignFedAvg, (Chen et al., 2020a) cannot allow multiple local SGD updates.

## 3.2 CASE 2: $z = +\infty$

When $z = +\infty$, the injected noise $\xi_z$ in the $z$-SignFedAvg is uniformly distributed on $[-1, 1]$. From Remark 1, we have learned that the gradient bias can vanish as long as the noise scale $\sigma$ is sufficiently large. To quantify this threshold, we need the following assumption which is a limit form of Assumption 2.

**Assumption 3.** *There exists a constant $Q_\infty \geq 0$ such that for any $x \in \mathbb{R}^d$, with probability 1,*

$$\|g_i(x) - \nabla f_i(x)\|_\infty \leq Q_\infty. \tag{8}$$

**Theorem 2.** *(Informal) Suppose that Assumption 1 and 3 hold. For $\gamma = \min\{n^{\frac{1}{2}}\tau^{-\frac{1}{2}}, L_{\max}^{-1}\}$, $\eta = \sigma$, $z = +\infty$, $E \leq n^{-\frac{3}{4}}\tau^{\frac{1}{4}}$ and $\sigma > E(G + Q_\infty)$ in Algorithm 1 we have*

$$\mathbb{E}\left[\frac{1}{\tau}\sum_{t=1}^{T}\sum_{s=1}^{E}\|\nabla f(\bar{x}_{t-1,s-1})\|^2\right] = \mathcal{O}\left((n\tau)^{-\frac{1}{2}}\right). \tag{9}$$

*However, if $\sigma \leq E(G+Q_\infty)$, there exists a problem instance for which Algorithm 1 cannot converge.*

**Remark 2.** *Note that Theorem 2 implies that $\infty$-SignFedAvg has a matching convergence rate as the uncompressed FedAvg. The reason why $\infty$-SignFedAvg cannot converge when $\sigma \leq E(G + Q_\infty)$ is simply that the uniform noise has a finite support and cannot always change the sign of gradients. For example, if $\sigma < A$ for some $A > 0$, then we have $\text{Sign}(x + \sigma\xi_\infty) = \text{Sign}(x)$ for any $x \geq A$.*

**Relationship to (Jin et al., 2020; Safaryan & Richtárik, 2021).** As mentioned in Remark 1, both the stochastic sign operators in (Jin et al., 2020; Safaryan & Richtárik, 2021) are equivalent to the sign operator injected by the uniform noise. Nevertheless, there are still two distinctions when compared with our $\infty$-SignFedAvg. First, while (Safaryan & Richtárik, 2021) shows their algorithm has a $\mathcal{O}(\tau^{-\frac{1}{4}})$ convergence rate, it is based on the $\ell_2$-norm of gradients and cannot imply the same rate as that in (9) (see Appendix A). Second, although (Safaryan & Richtárik, 2021) does not need Assumption 3, it relies on an input-dependent noise scale which, unfortunately, often slows the algorithm convergence in practice especially when the problem dimension is large.

More theoretical results and proofs are relegated to Appendix B and C. Below, we have two more remarks.

**Remark 3.** *(Bounded minibatch gradient noise) While both Assumption 2 and 3 are slightly stronger than the commonly used second-order condition on the minibatch gradient noise, they are still justifiable since unbounded minibatch gradient noise is rarely to happen in practice.*

Table 1: Comparison of Case 1 and Case 2.

| Case | Convergence rate | Threshold on $\sigma$ | Assumption on gradient noise |
|---|---|---|---|
| $z < +\infty$ | $\mathcal{O}(\tau^{-\frac{z}{2z+1}})$ | $\widetilde{\mathcal{O}}\left(\left(\frac{Q_z}{G} + G^{4z}\right)^{\frac{1}{4z}}\right)$ | Assumption 2 |
| $z = +\infty$ | $\mathcal{O}(\tau^{-\frac{1}{2}})$ | $\widetilde{\mathcal{O}}(Q_\infty + G)$ | Assumption 3 |

**Remark 4.** *(Minibatch gradient noise works as noise perturbation) When the minibatch gradient is used as the input of the sign operator in (2), the minibatch gradient noise itself may function as the perturbation noise. In particular, as shown in (Chen et al., 2020b) the minibatch gradient noise approximately follows a symmetric distribution. Therefore, in practice, one may not need to inject as large noises as suggested by Theorem 2 since the minibatch gradient noise can also help mitigate the bias due to sign-based compression. This also explains why a small noise scale is sufficient for $z$-SignFedAvg to achieve good performance in the experiment section.*

## 3.3 COMPARISON OF CASE 1 AND CASE 2

We summarize the results of Case 1 and Case 2 in Table 1, where $\widetilde{O}(\cdot)$ hides some constants that do not affect the comparison. Especially, we can see that when the mini-batch gradient noise has a long tail such that $Q_z/G \ll Q_\infty^{4z}$, Case 1 requires a less amount of noise than Case 2 for guaranteeing convergence. Despite of the difference in theory, we will see in Section 4 that $z$-SignFedAvg under Case 1 and Case 2 have almost the same behavior in practice.

## 3.4 IMPLICATION ON DIFFERENTIALLY PRIVATE FEDERATED LEARNING (DP-FL)

Beyond the convergence issue, we remark that adding Gaussian noise to the local gradients is also a common practice for privacy protection, especially in DP-FL (Geyer et al., 2017; Agarwal et al., 2021; 2018). With this observation, it is straightforward to propose a differentially-private variant

of $z$-SignFedAvg, which we term DP-SignFedAvg. More details and comparison results between DP-SignFedAvg and the uncompressed DP-FedAvg (Geyer et al., 2017; Kairouz et al., 2021) under different privacy budgets are given in Appendix F.

## 4 EXPERIMENTS

In this section, we present the experiment results on both synthetic and real problems, and all the figures in this section are obtained by 10 independent runs and are visualized in the form of mean±std.

**Noise scale as a hyperparameter.** Although we explicitly characterize how the performance of $z$-SignFedAvg depends on the noise scale $\sigma$ in the previous section, we treat $\sigma$ as a tunable hyperparameter in the experiments. This is because, on one hand, the theoretical lower bound for $\sigma$ are difficult to compute since it is impossible to access the moment condition of the minibatch gradient noise. On the other hand, as we have discussed in Remark 4, owing to the presence of the minibatch gradient noise, we can use a much smaller noise scale than the theoretical one in practice.

Aside from the experiments presented in this section, we also compare our algorithm to another popular family of unbiased stochastic compressed FL algorithms, namely, the QSGD in (Alistarh et al., 2017) and FedPAQ in (Reisizadeh et al., 2020). For detailed results, we refer readers to Appendix E.

### 4.1 A SIMPLE CONSENSUS PROBLEM

In this section, we verify our theoretical results in Section 3 by considering the simple consensus problem with 10 clients: $\min_{x \in \mathbb{R}^d} \frac{1}{2} \sum_{i=1}^{10} \|x - y_i\|^2$, where $y_1, ..., y_{10} \in \mathbb{R}^d$ are generated using i.i.d standard Gaussian distribution, and $d$ is the problem dimension. We implemented the following algorithms: GD (Gradient descent), Sto-SignSGD (Safaryan & Richtárik, 2021), SignSGD (Algorithm 1 with $z = 1$, $E = 1$ and $\sigma = 0$), 1-SignSGD (Algorithm 1 with $z = 1$ and $E = 1$.), $\infty$-SignSGD (Algorithm 1 with $z = +\infty$ and $E = 1$). For all the algorithms, we considered the full gradient (no mini-batch SGD), and used the same stepsize $0.01$ and initialization by a zero vector.

**Results.** As we can see from Figure 1, the vanilla SignSGD fails to converge to the optimal solution whereas the others can. Besides, 1-SignSGD and $\infty$-SignSGD have roughly the same convergence speed which is slightly slower than the uncompressed GD. It is also observed that the input-dependent noise scale adopted by (Safaryan & Richtárik, 2021) could slow the convergence when the problem dimension is high, as discussed in Section 3.2.

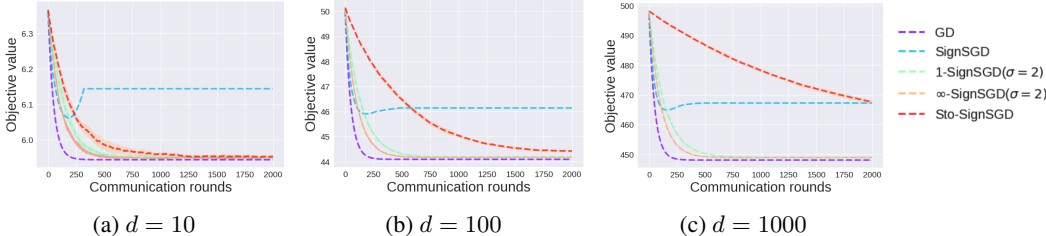

|          |          |          |
|:--------:|:--------:|:--------:|
| (a) $d = 10$ | (b) $d = 100$ | (c) $d = 1000$ |

Figure 1: Performance of tested algorithms under different problem dimension.

Figure 2 displays the results of 1-SignSGD and $\infty$-SignSGD with various noise scales. We can see that there is a clear bias-variance trade-off for different noise scales and it corroborates our analysis after Theorem 1. It is also worth mentioning that the best choice of $\sigma$ for Algorithm 1 shown in Figure 2 is much smaller than the one predicted by the theorems.

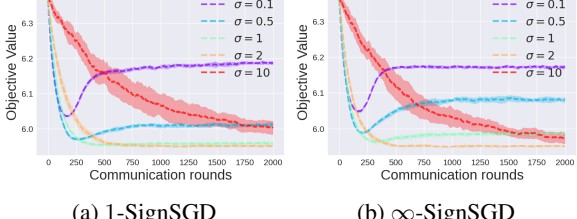

|          |          |
|:--------:|:--------:|
| (a) 1-SignSGD | (b) $\infty$-SignSGD |

Figure 2: $z$-SignSGD under various noise scales.

### 4.2 $z$-SIGNSGD ON NON-I.I.D MNIST

In this section, we consider an extremely non-i.i.d setting with the MNIST dataset (Deng, 2012). Specifically, we split the dataset into 10 parts based on the labels and each client has the data of one digit only. A simple two-layer convolutional neural network (CNN) from Pytorch tutorial (Paszke et al., 2017) was used for the learning task. The following algorithms were implemented: SGDwM (Distributed SGD (Ghadimi & Lan, 2013) with momentum), EF-SignSGDwM (Distributed

SignSGD with error-feedback and momentum (Karimireddy et al., 2019; Vogels et al., 2019)), and Sto-SignSGDwM (Sto-SignSGD with momentum (Safaryan & Richtárik, 2021)). For each of the algorithms, we selected its best hyperparameters, including the stepsize, momentum coefficient and the noise scale, via grid search (see Appendix D.1).

**Results.** One can observe from Figure 3a-3b that again the vanilla SignSGD does not converge well. The proposed 1-SignSGD and $\infty$-SignSGD clearly outperform the existing EF-SignSGDwM and Sto-SignSGDw, and perform closely to the uncompressed SGDwM. The reason for the slow convergence of Sto-SignSGDw is that the injected noise is too large due to the input-dependent noise scale. Figure 3c further displays the testing accuracy of all methods versus the accumulated number of bits transmitted from the clients to the server. One can see that the proposed algorithms achieve the state-of-the-art performance on this task. More results for 1-SignSGD and $\infty$-SignSGD under different noise scales are presented in Appendix D.1.

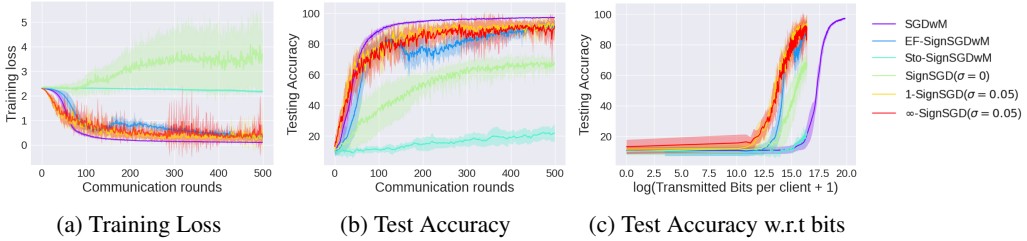

| (a) Training Loss | (b) Test Accuracy | (c) Test Accuracy w.r.t bits |

Figure 3: Performance of various SignSGD algorithms on non-i.i.d MNIST.

### 4.3 $z$-SIGNFEDAVG ON EMNIST AND CIFAR-10

In this section, we evaluate the performance of our proposed $z$-SignFedAvg on two classical datasets: EMNIST(Cohen et al., 2017) and CIFAR-10 (Krizhevsky & Hinton, 2010). In particular, the proposed $z$-SignFedAvg with $z = 1$ and $z = \infty$ are benchmarked against the uncompressed FedAvg (McMahan et al., 2017; Yu et al., 2019). Since 1-SignFedAvg and $\infty$-SignFedAvg behave similarly, we only report the results of 1-SignFedAvg in this section and relegate the others to Appendix D.2. For EMNIST, we use the same 2-layer CNN as the one in Section 4.2. For CIFAR-10, we used the ResNet18 (He et al., 2016) with group normalization (Wu & He, 2018).

**Settings.** For both the experiments on EMNIST and CIFAR-10, we followed a setting similar to (Reddi et al., 2020). We also considered the scenario with partial client participation. For the EMNIST dataset, there are 3579 clients in total and 100 clients were uniformly sampled in each communication round to upload their compressed gradients. For the CIFAR-10 dataset, the training samples are partitioned among 100 clients, and each client has an associated multinomial distribution over labels drawn from a symmetric Dirichlet distribution with parameter 1. In each communication round, 10 out of 100 clients were uniformly sampled. The same noise scales for 1-SignFedAvg and $\infty$-SignFedAvg were used: $\sigma = 0.01$ for EMNIST and $\sigma = 0.0005$ for CIFAR-10. More details about the hyperparameters are referred to Appendix D.2.

**Results.** We can see from Figure 4 and Figure 5 that both uncompressed FedAvg and 1-SignFedAvg can benefit from multiple local SGD steps. More surprisingly, 1-SignFedAvg can even outperform the uncompressed FedAvg. This is probably because the EMNIST dataset is less heterogeneous than the one we used in Section 4.2. The results on the performance of 1-SignFedAvg and $\infty$-SignFedAvg under various choices of noise scales are relegated to Appendix D.2, which are also consistent with our theoretical claims in Section 3.

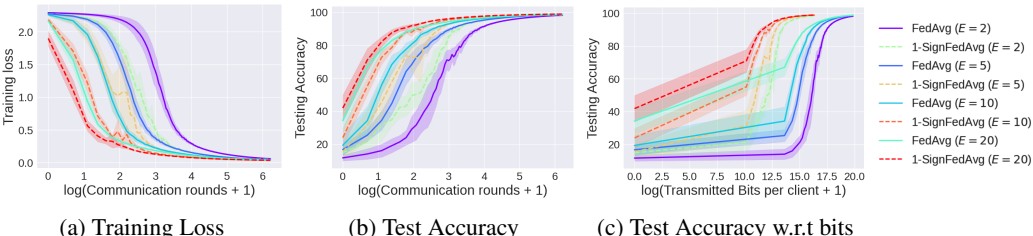

| (a) Training Loss | (b) Test Accuracy | (c) Test Accuracy w.r.t bits |

Figure 4: Performance of FedAvg and 1-SignFedAvg on the EMNIST dataset.

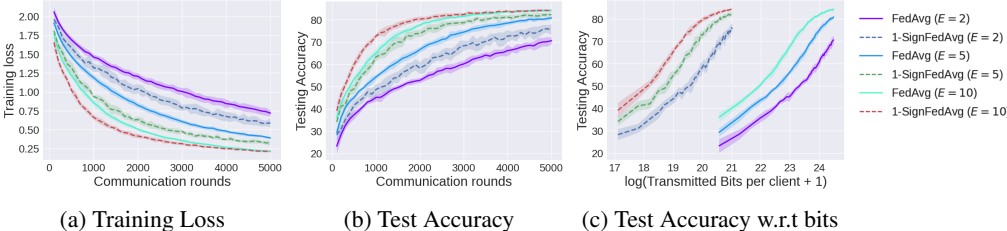

(a) Training Loss        (b) Test Accuracy      (c) Test Accuracy w.r.t bits

Figure 5: Performance of FedAvg and 1-SignFedAvg on the CIFAR-10 dataset.

### 4.4 PLATEAU CRITERION FOR TUNING THE NOISE SCALE

From previous experiments, we have learned that the noise scale $\sigma$ has to be properly chosen for the algorithm to perform well. However, it could be time-consuming to select the optimal noise scale via grid search. Therefore, here we introduce a simple yet useful strategy that can tune the noise scale adaptively during the training process. Figure 2 indicates that the noise scale should plays a similar role as the stepsize when training a neural network: Small noise scale leads to fast convergence at the beginning, while large noise scale guarantees a better final performance. This suggests that we should use an increasing noise scale during the optimization process. We can also see this from Corollary 1 because that the noise scale $\sigma$ is proportional to $\tau$. Besides, it has been shown that the gradients of neural network tend to be sparser during the training process (Karimireddy et al., 2019). Therefore, as studied in (Isik & Weissman, 2022), from the rate-distortion theoretic aspect, the noise scale should be increasing as the compression becomes more aggressive. Motivated by all of these insights, we propose the following Plateau criterion for adapting the noise scale.

**Plateau criterion.** We denote a few parameters $\sigma_{\text{bound}} \geq \sigma_{\text{init}} > 0$, $\kappa \in \mathbb{Z}_+$, $\beta > 0$. We first start Algorithm 1 with a small noise scale $\sigma_{\text{init}}$, i.e., $\sigma = \sigma_{\text{init}}$, and then update the noise scale via $\sigma = \beta\sigma$, where $\beta \in [1.5, 2]$, whenever the objective function stops improving for $\kappa$ communication rounds. We stop updating $\sigma$ if it has already been greater than a relatively large number $\sigma_{\text{bound}}$.

**Results.** We demonstrate the efficacy of the Plateau criterion by comparing the performance of 1-SignSGD/1-SignFedAvg with the optimal noise scale found in previous experiments and the ones with Plateau criterion. Figure 6 shows the results under the three different settings used in Section 4.2 and 4.3. We can see that, the Plateau criterion could results in a slower convergence speed than the optimal noise scale in the middle phase of optimization, because it requires some time for the algorithm to adapt to a suitable noise scale. But eventually it can lead to the same objective value obtained by using the optimal noise scale. For more details like the hyperparameters for Plateau criterion and the evolution of noise scale, we refer readers to Appendix D.3.

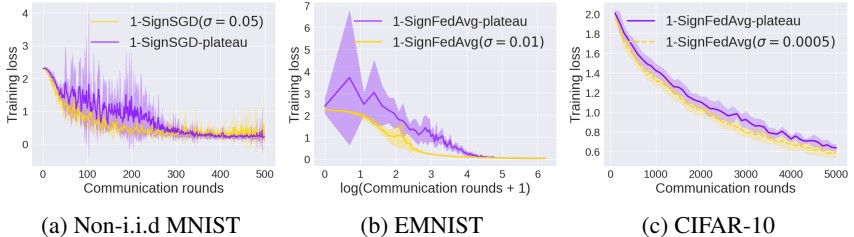

(a) Non-i.i.d MNIST        (b) EMNIST        (c) CIFAR-10

Figure 6: Evaluating the efficacy of Plateau criterion on three different datasets.

## 5 CONCLUSION

In this work, we have proposed the $z$-SignFedAvg: a FedAvg-type algorithm with the stochastic sign-based compression. Thanks to the novel noisy perturbation scheme in Section 2, the proposed $z$-SignFedAvg provides a unified viewpoint to the existing sign-based methods as well as a general framework for convergence rate analysis. Through both theoretical analyses and empirical experiments, we have shown that the $z$-SignFedAvg can perform nearly the same, sometimes even better, than the uncompressed FedAvg and enjoy a significant reduction in the number of bits transmitted from clients to the server. As a final remark, the stochastic sign-based compression proposed in this work can be of independent interest and can be conveniently combined with other adaptive FL algorithms or gradient sparsification techniques such as those in (Karimireddy et al., 2020; Reddi et al., 2020; Basu et al., 2019), to further improve the communication efficiency.

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

## APPENDIX

### A  COMPARISON WITH EXISTING STOCHASTIC SIGN-BASED METHODS

Table 2 summarizes the key features of a few representative stochastic sign-based methods and the proposed $z$-SignFedAvg, including the convergence rate, metric used in the convergence rate analysis, extra assumptions other than A.1-A.3 in Assumption 1, and whether the algorithm can achieve linear speedup and allow multiple local SGD steps.

For communication complexity, we focus on the uplink communication cost, i.e., the number of bits transmitted from the clients to the server in each communication round. We assume that all the uncompressed algorithms use 32 bits to represent a single float number as it is the most common setting in Tensorflow (Abadi et al., 2016a) and Pytorch (Paszke et al., 2017).

While most of the existing methods use the squared $\ell_2$-norm of gradients as the convergence metric, the work (Safaryan & Richtárik, 2021) adopts the $\ell_2$-norm of gradients. The work (Chen et al., 2020a) uses a convergence metric mixed with squared $\ell_2$-norm and $\ell_1$-norm of gradients due to the compression in both uplink and downlink .

Among the works in Table 2, the setting considered by (Safaryan & Richtárik, 2021) is closest to ours. (Safaryan & Richtárik, 2021) proposed an algorithm that can achieve the convergence rate $\mathcal{O}(\tau^{-\frac{1}{4}})$ with the $\ell_2$-norm of gradients as the metric. We remark that this is inferior to the convergence rate $\mathcal{O}(\tau^{-\frac{1}{2}})$ with the squared $\ell_2$-norm as the metric. To illustrate this point, we denote a series of vector as $\{\alpha_1, ..., \alpha_\tau, ...\}$ with $\alpha_i \in \mathbb{R}^d$. If now

$$\frac{1}{\tau}\sum_{i=1}^{\tau}\|\alpha_i\| = \mathcal{O}(\tau^{-\frac{1}{4}}), \tag{10}$$

in the worst case, we can only guarantee that

$$\frac{1}{\tau}\sum_{i=1}^{\tau}\|\alpha_i\|^2 \leq \tau\left(\frac{1}{\tau}\sum_{i=1}^{\tau}\|\alpha_i\|\right)^2 = \mathcal{O}(\tau^{\frac{1}{2}}). \tag{11}$$

As a simple example, the equality in (11) holds if and only if there is exactly one non-zero term in $\{\alpha_1, ..., \alpha_\tau\}$.

On the contrary, if it holds that

$$\frac{1}{\tau}\sum_{i=1}^{\tau}\|\alpha_i\|^2 = \mathcal{O}(\tau^{-\frac{1}{2}}), \tag{12}$$

| Algorithm | Convergence rate / metric | Num. of bits per commun. round | Extra Assumptions? | Can achieve linear speedup? | Can allow multiple local steps? |
|---|---|---|---|---|---|
| (Ghadimi & Lan, 2013) | $\mathcal{O}(\tau^{-\frac{1}{2}})$ squared $\ell_2$ | $32d$ | No | ✓ | ✗ |
| (McMahan et al., 2017) (Yu et al., 2019) | $\mathcal{O}(\tau^{-\frac{1}{2}})$ squared $\ell_2$ | $32d$ | • Bounded gradient | ✓ | ✓ |
| (Karimireddy et al., 2019) | $\mathcal{O}(\tau^{-\frac{1}{2}} + d^2\tau^{-1})$ squared $\ell_2$ | $d + 32$ | • Bounded gradient | ✗ | ✗ |
| (Safaryan & Richtárik, 2021) | $\mathcal{O}(\tau^{-\frac{1}{4}})$ $\ell_2$ | $d$ | No | ✗ | ✗ |
| (Jin et al., 2020) | $\mathcal{O}(\tau^{-\frac{1}{4}})$ squared $\ell_2$ | $d$ | • Bounded gradient • n is an odd number | ✗ | ✗ |
| (Chen et al., 2020a) | $\mathcal{O}(\tau^{-\frac{1}{4}})$ mixed | $d$ | • Bounded gradient • n is an odd number | ✗ | ✗ |
| (Alistarh et al., 2017) | $\mathcal{O}(\tau^{-\frac{1}{2}})$ squared $\ell_2$ | $\approx sd + 32$ | No | ✓ | ✗ |
| (Haddadpour et al., 2021) | $\mathcal{O}(\tau^{-\frac{1}{2}})$ squared $\ell_2$ | $\approx sd + 32$ | • Bounded gradient dissimilarity | ✓ | ✓ |
| 1-SignFedAvg (ALG. 1) **This work** | $\mathcal{O}(\tau^{-\frac{1}{3}})$ squared $\ell_2$ | $d$ | • Bounded gradient • Bounded 6th moment of gradient noise | ✓ | ✓ |
| $\infty$-SignFedAvg (ALG. 1) **This work** | $\mathcal{O}(\tau^{-\frac{1}{2}})$ squared $\ell_2$ | $d$ | • Bounded gradient • Bounded support of gradient noise | ✓ | ✓ |

Table 2: Summary of representative stochastic sign-based methods.

then we have

$$\frac{1}{\tau}\sum_{i=1}^{\tau}\|\alpha_i\| \leq \sqrt{\frac{1}{\tau}\sum_{i=1}^{\tau}\|\alpha_i\|^2} = \mathcal{O}(\tau^{-\frac{1}{4}}). \tag{13}$$

Thus, the convergence results in (Safaryan & Richtárik, 2021) cannot imply the rate in Theorem 2. Besides, the algorithm in (Safaryan & Richtárik, 2021) is equivalent to our Algorithm 1 with $z = \infty$, $E = 1$ and $\sigma = \|g_{t-1,s}^i\|$. This input-dependent noise scale is linearly increasing w.r.t the problem dimension and is too conservative for practical applications. From Figure 1 and Figure 3, we have already seen that this input-dependent noise scale could result in an extremely slow convergence for high-dimensional problems.

Except for the previous sign-based compression methods, another type of compressed FL algorithms, such as (Alistarh et al., 2017) and (Haddadpour et al., 2021), adopt a unified unbiased compressor $Q(\cdot)$ that satisfies $\mathbb{E}[\|Q(x) - x\|^2] \leq C\|x\|^2$ for some constant $C > 0$. We remark that such property is not fulfilled by any of the existing sign-based compressors. Thus, the theoretical results therein cannot be applied to sign-based methods. A specific example of such unbiased compressor is described below.

**Definition 2** (Unbiased quantizer). *For any variable $x \in \mathbb{R}^d$, the unbiased quantizer $Q(\cdot) : \mathbb{R}^d \to \mathbb{R}^d$ is defined as below*

$$Q(x) = \|x\|_2 \cdot \begin{bmatrix} \text{Sign}(x_1)\xi(x_1, s) \\ \text{Sign}(x_2)\xi(x_2, s) \\ \vdots \\ \text{Sign}(x_d)\xi(x_d, s) \end{bmatrix} \tag{14}$$

*where $\xi(x_i, s)$ is a random variable taking on value $\frac{l+1}{s}$ with probability $\frac{|x_i|}{\|x\|_2} s - l$ and $\frac{l}{s}$ otherwise. Here, the tuning parameter $s$ corresponds to the number of quantization levels and $l \in [0, s)$ is an integer such that $\frac{|x_i|}{\|x\|_2} \in [l/s, l + 1/s)$.*

In Table 2, we assume both (Alistarh et al., 2017) and (Haddadpour et al., 2021) adopt the quantizer in (14). Generally speaking, this type of unbiased quantization usually requires much more bits than sign-based compression to obtain a good performance, which is also verified empirically in Appendix E. It is also worthwhile to mention that the FedPAQ in (Reisizadeh et al., 2020) and the FedCOM in (Haddadpour et al., 2021) are equivalent in algorithm, but only the latter one considers the heterogeneous scenario theoretically.

## B  DETAILED THEORETICAL RESULTS

We first state the result on the limit of $z$-distribution.

**Lemma 2.** *The $z$-distribution weakly converges to uniform distribution on $[-1, 1]$ when $z \to +\infty$.*

The following corollary is the formal version of Corollary 1.

**Corollary 2** (Formal version of Corollary 1). *For $\gamma = \min\{n^{\frac{z}{2z+1}} \tau^{-\frac{z+1}{2z+1}}, \frac{1}{L_{\max}}\}$ and $\sigma = (n\tau)^{\frac{1}{4z+2}}$ in Theorem 1, we have*

$$
\mathbb{E}\left[\frac{1}{\tau} \sum_{t=1}^{T} \sum_{s=1}^{E} \|\nabla f(\bar{x}_{t-1,s-1})\|^2\right] \leq \frac{2\mathbb{E}[f(x_0) - f^*]}{(n\tau)^{\frac{z}{2z+1}}} + \frac{\zeta^2 L_{\max}}{(n\tau)^{\frac{z+1}{2z+1}}} + \frac{4(E-1)En^{\frac{2z}{2z+1}} L_{\max}^2 (\zeta^2 + G^2)}{3\tau^{\frac{2z+2}{2z+1}}}
$$

$$
+ \frac{2^{2z+1} E^{2z} \sqrt{Q_z + G^{4z+2}} G}{\sqrt{2}(2z+1)(n\tau)^{\frac{z}{2z+1}}} + \frac{2^{4z} E^{4z+1} (Q_z + G^{4z+2}) L_{\max}}{2(2z+1)^2 n^{\frac{z}{2z+1}} \tau^{\frac{3z+1}{2z+1}}}
$$

$$
+ \frac{4\eta_z^2 \sum_{j=1}^{d} L_j}{E(n\tau)^{\frac{z}{2z+1}}}. \tag{15}
$$

*Furthermore, if $E \leq n^{-\frac{3z}{4z+2}} \tau^{\frac{z+2}{4z+2}}$, the upper bound above becomes*

$$
\mathbb{E}\left[\frac{1}{\tau} \sum_{t=1}^{T} \sum_{s=1}^{E} \|\nabla f(\bar{x}_{t-1,s-1})\|^2\right] \leq \frac{2\mathbb{E}[f(x_0) - f^*]}{(n\tau)^{\frac{z}{2z+1}}} + \frac{\zeta^2 L_{\max}}{(n\tau)^{\frac{z+1}{2z+1}}} + \frac{4 L_{\max}^2 (\zeta^2 + G^2)}{3(n\tau)^{\frac{z}{2z+1}}}
$$

$$
+ \frac{2^{2z+1} E^{2z} \sqrt{Q_z + G^{4z+2}} G}{\sqrt{2}(2z+1)(n\tau)^{\frac{z}{2z+1}}} + \frac{2^{4z} E^{4z+1} (Q_z + G^{4z+2}) L_{\max}}{2(2z+1)^2 n^{\frac{z}{2z+1}} \tau^{\frac{3z+1}{2z+1}}}
$$

$$
+ \frac{4\eta_z^2 \sum_{j=1}^{d} L_j}{E(n\tau)^{\frac{z}{2z+1}}}. \tag{16}
$$

.

The formal version of Theorem 2 is given below.

**Theorem 3** (Formal version of Theorem 2). *Suppose that Assumption 1 and 3 hold. For $\gamma \leq \frac{1}{L_{\max}}$, $\eta = \sigma$, $z = +\infty$ and $\sigma > E(G + Q_\infty)$ in Algorithm 1, we have*

$$
\mathbb{E}\left[\frac{1}{TE} \sum_{t=1}^{T} \sum_{s=1}^{E} \|\nabla f(\bar{x}_{t-1,s-1})\|^2\right] \leq \underbrace{\frac{2\mathbb{E}[f(x_0) - f^*]}{TE\gamma} + \frac{\gamma\zeta^2 L_{\max}}{n} + \frac{4\gamma^2 (E-1)EL_{\max}^2(\zeta^2 + G^2)}{3}}_{\text{Standard terms in FedAvg}}
$$

$$
+ \underbrace{\frac{4\gamma\sigma^2 \sum_{j=1}^{d} L_j}{En}}_{\text{Variance term}}. \tag{17}
$$

*Otherwise, if $\sigma \leq E(G + Q_\infty)$, there exists a problem instance for which the algorithm cannot converge. If we further choose $\gamma = \min\{n^{\frac{1}{2}}\tau^{-\frac{1}{2}}, \frac{1}{L_{\max}}\}$, we have*

$$\mathbb{E}\left[\frac{1}{\tau}\sum_{t=1}^{T}\sum_{s=1}^{E}\|\nabla f(\bar{x}_{t-1,s-1})\|^2\right] \leq \frac{2\mathbb{E}[f(x_0) - f^*]}{(n\tau)^{\frac{1}{2}}} + \frac{\zeta^2 L_{\max}}{(n\tau)^{\frac{1}{2}}} + \frac{4(E-1)EnL_{\max}^2\left(\zeta^2 + G^2\right)}{3\tau}$$

$$+ \frac{4\sigma^2\sum_{j=1}^{d}L_j}{E(n\tau)^{\frac{1}{2}}}. \tag{18}$$

*Furthermore, if $E \leq n^{-\frac{3}{4}}\tau^{\frac{1}{4}}$, the upper bound above becomes*

$$\mathbb{E}\left[\frac{1}{\tau}\sum_{t=1}^{T}\sum_{s=1}^{E}\|\nabla f(\bar{x}_{t-1,s-1})\|^2\right] \leq \frac{2\mathbb{E}[f(x_0) - f^*]}{(n\tau)^{\frac{1}{2}}} + \frac{\zeta^2 L_{\max}}{(n\tau)^{\frac{1}{2}}} + \frac{4L_{\max}^2\left(\zeta^2 + G^2\right)}{3(n\tau)^{\frac{1}{2}}}$$

$$+ \frac{4\sigma^2\sum_{j=1}^{d}L_j}{E(n\tau)^{\frac{1}{2}}}, \tag{19}$$

*which recovers the convergence result of the uncompressed FedAvg algorithm (Yu et al., 2019).*

In particular, since the third term in the RHS of (18) is $\mathcal{O}(E^2 n\tau^{-1})$, hence when $E \leq n^{-\frac{3}{4}}\tau^{\frac{1}{4}}$, this term becomes $\mathcal{O}((n\tau)^{-\frac{1}{2}})$.

## C  PROOFS

### C.1  PROOF OF LEMMA 1

We first state a useful inequality on the c.d.f of the $z$-distribution:

**Lemma 3.** *For any $x \in \mathbb{R}$, it holds that*

$$|x| - \frac{|x|^{2z+1}}{2(2z+1)} \leq |\Psi_z(x)| \leq |x|, \tag{20}$$

*where*

$$\Psi_z(x) \overset{\text{def.}}{=} \int_0^x e^{-\frac{t^{2z}}{2}}\,dt.$$

Similar to the sign operator, for any vector $x = [x(1), ..., x(d)]^\top \in \mathbb{R}^d$, we define

$$\Psi_z(x) = [\Psi_z(x(1)), ..., \Psi_z(x(d))]^\top.$$

With the presence of Lemma 3, we have

$$\|\eta_z \sigma \mathbb{E}[\text{Sign}(\mathbf{x} + \sigma\xi_{\mathbf{z}})] - x\|^2 = \left\|x - \sigma\Psi_z\left(\frac{x}{\sigma}\right)\right\|^2 = \sum_{j=1}^{d}\left(x(j) - \sigma\Psi_z\left(\frac{x(j)}{\sigma}\right)\right)^2$$

$$\leq \sum_{j=1}^{d}\frac{(x(j))^{4z+2}}{4(2z+1)^2\sigma^{4z}} = \frac{\|x\|_{4z+2}^{4z+2}}{4(2z+1)^2\sigma^{4z}}. \tag{21}$$

*Proof of Lemma 3.* Without loss of generality, we consider $x \geq 0$. First,

$$\int_0^x e^{-\frac{t^{2z}}{2}}\,dt \leq \int_0^x 1\,dt \leq x. \tag{22}$$

Now we define $F(x) \overset{\text{def.}}{=} \int_0^x e^{-\frac{t^{2z}}{2}}\,dt - x + \frac{x^{2z+1}}{2(2z+1)}$. Note that $F(0) = 0$. Then, it suffices to show $F(x) \geq 0$ by

$$F'(x) = e^{-\frac{x^{2z}}{2}} - 1 + \frac{x^{2z}}{2} \geq 0. \tag{23}$$

It is true since the inequality $e^{-t} - 1 + t \geq 0$ for any $t \geq 0$. $\qquad\square$

### C.2 PROOF OF LEMMA 2

Now we denote the p.d.f of the uniform distribution as

$$p_\infty(x) = \begin{cases} \frac{1}{2} & |x| \leq 1, \\ 0 & |x| > 1. \end{cases} \tag{24}$$

Without loss of generality, for any $x > 1$ and $z \in \mathbb{Z}_+$, we have

$$\left| \int_{-\infty}^x \frac{1}{2\eta_z} e^{-\frac{t^{2z}}{2}} dt - \int_{-\infty}^x p_\infty(t) dt \right| = \left| \int_0^x \left( \frac{1}{2\eta_z} e^{-\frac{t^{2z}}{2}} - p_\infty(t) \right) dt \right|$$

$$\leq \int_0^1 \left| \frac{1}{2\eta_z} e^{-\frac{t^{2z}}{2}} - \frac{1}{2} \right| dt + \int_1^x \frac{1}{2\eta_z} e^{-\frac{t^{2z}}{2}} dt. \tag{25}$$

For any $0 < \epsilon < \min\{1, x-1\}$, we have

$$\int_0^1 \left| \frac{1}{2\eta_z} e^{-\frac{t^{2z}}{2}} - \frac{1}{2} \right| dt = \int_0^{1-\epsilon} \left| \frac{1}{2\eta_z} e^{-\frac{t^{2z}}{2}} - \frac{1}{2} \right| dt + \int_{1-\epsilon}^1 \left| \frac{1}{2\eta_z} e^{-\frac{t^{2z}}{2}} - \frac{1}{2} \right| dt$$

$$\leq \left| \frac{1}{2\eta_z} e^{-\frac{(1-\epsilon)^{2z}}{2}} - \frac{1}{2} \right| + \epsilon. \tag{26}$$

Since $\lim_{z\to\infty} \frac{1}{2\eta_z} = \lim_{z\to\infty} \frac{z}{2^{\frac{1}{2z}} \Gamma(\frac{1}{2z})} = \frac{1}{2}$ and $\lim_{z\to\infty} e^{-\frac{(1-\epsilon)^{2z}}{2}} = 1$, there exists an integer $Z_1 > 0$ such that if $z > Z_1$, we have

$$\left| \frac{1}{2\eta_z} e^{-\frac{(1-\epsilon)^{2z}}{2}} - \frac{1}{2} \right| \leq \epsilon.$$

Similarly, we have

$$\int_1^x \frac{1}{2\eta_z} e^{-\frac{t^{2z}}{2}} dt = \int_1^{1+\epsilon} \frac{1}{2\eta_z} e^{-\frac{t^{2z}}{2}} dt + \int_{1+\epsilon}^x \frac{1}{2\eta_z} e^{-\frac{t^{2z}}{2}} dt$$

$$\leq \epsilon + \frac{1}{2\eta_z} e^{-\frac{(1+\epsilon)^{2z}}{2}} (x - 1 - \epsilon). \tag{27}$$

Since $\lim_{z\to\infty} e^{-\frac{(1+\epsilon)^{2z}}{2}} = 0$, there exists an integer $Z_2 > 0$ such that if $z > Z_2$, we have

$$\int_1^x \frac{1}{2\eta_z} e^{-\frac{t^{2z}}{2}} dt \leq \epsilon. \tag{28}$$

In all, for any $0 < \epsilon < \min\{1, x-1\}$, if $z$ is sufficiently large, we have

$$\left| \int_{-\infty}^x \frac{1}{2\eta_z} e^{-\frac{t^{2z}}{2}} dt - \int_{-\infty}^x p_\infty(t) dt \right| \leq 4\epsilon. \tag{29}$$

Taking $\epsilon \to 0$ and $z \to \infty$, we have

$$\lim_{z\to\infty} \left| \int_{-\infty}^x \frac{1}{2\eta_z} e^{-\frac{t^{2z}}{2}} dt - \int_{-\infty}^x p_\infty(t) dt \right| = 0. \tag{30}$$

### C.3 PROOF OF THEOREM 1

We denote the aggregated update $\bar{x}_t = \bar{x}_{t-1,E}$. First, we state two technical lemmas:

**Lemma 4.** *Suppose that Assumption 1 and 2 hold. For the t-th ($1 \leq t \leq T$) communication round in Algorithm 1, if $\eta = \eta_z \sigma$ and $z < +\infty$, we have*

$$\mathbb{E}[f(x_t) - f(\bar{x}_t)] \leq \frac{\gamma^2 2^{2z} E^{2z+1} \sqrt{Q_z + G^{4z+2}} G}{\sqrt{2}(2z+1)\sigma^{2z}} + \frac{\gamma^2 2^{4z} E^{4z+2}(Q_z + G^{4z+2}) L_{\max}}{4(2z+1)^2 \sigma^{4z}}$$

$$+ \frac{2\eta_z^2 \gamma^2 \sigma^2 \sum_{j=1}^d L_j}{n}. \tag{31}$$

**Lemma 5.** *Suppose that Assumption 1 hold. For the $t$-th ($1 \leq t \leq T$) communication round in Algorithm 1, if $\gamma \leq \frac{1}{L_{\max}}$, we have*

$$\mathbb{E}[f(\bar{x}_t) - f(x_{t-1})] \leq -\frac{\gamma}{2} \sum_{s=1}^{E} \|\nabla f(\bar{x}_{t-1,s-1})\|^2 + \frac{E\gamma^2\zeta^2 L_{\max}}{2n} + \frac{2\gamma^3(E-1)E^2 L_{\max}^2(\zeta^2 + G^2)}{3}.$$
(32)

By combining Lemma 4 and Lemma 5, we have

$$\mathbb{E}[f(x_t) - f(x_{t-1})] = \mathbb{E}[f(x_t) - f(\bar{x}_t)] + E[f(\bar{x}_t) - f(x_{t-1})]$$

$$\leq -\frac{\gamma}{2} \sum_{s=1}^{E} \|\nabla f(\bar{x}_{t-1,s-1})\|^2 + \frac{E\gamma^2\zeta^2 L_{\max}}{2n} + \frac{2\gamma^3(E-1)E^2 L_{\max}^2(\zeta^2 + G^2)}{3}$$

$$+ \frac{\gamma 2^{2z} E^{2z+1}\sqrt{Q_z + G^{4z+2}}G}{\sqrt{2}(2z+1)\sigma^{2z}} + \frac{\gamma^2 2^{4z} E^{4z+2}(Q_z + G^{4z+2})L_{\max}}{4(2z+1)^2\sigma^{4z}}$$

$$+ \frac{2\eta_z^2\gamma^2\sigma^2 \sum_{j=1}^{d} L_j}{n}.$$
(33)

Rearranging the inequality (33), we have

$$\frac{1}{E} \sum_{s=1}^{E} \|\nabla f(\bar{x}_{t-1,s-1})\|^2 \leq \frac{2\mathbb{E}[f(x_{t-1}) - f(x_t)]}{E\gamma} + \frac{\gamma\zeta^2 L_{\max}}{n} + \frac{4\gamma^2(E-1)EL_{\max}^2(\zeta^2 + G^2)}{3}$$

$$+ \frac{2^{2z+1} E^{2z}\sqrt{Q_z + G^{4z+2}}G}{\sqrt{2}(2z+1)\sigma^{2z}} + \frac{\gamma 2^{4z} E^{4z+1}(Q_z + G^{4z+2})L_{\max}}{2(2z+1)^2\sigma^{4z}}$$

$$+ \frac{4\eta_z^2\gamma\sigma^2 \sum_{j=1}^{d} L_j}{En}.$$
(34)

Finally, by a telescopic sum, we obtain

$$\mathbb{E}\left[\frac{1}{TE} \sum_{t=1}^{T} \sum_{s=1}^{E} \|\nabla f(\bar{x}_{t-1,s-1})\|^2\right] \leq \frac{2\mathbb{E}[f(x_0) - f^*]}{TE\gamma} + \frac{\gamma\zeta^2 L_{\max}}{n} + \frac{4\gamma^2(E-1)EL_{\max}^2(\zeta^2 + G^2)}{3}$$

$$+ \frac{2^{2z+1} E^{2z}\sqrt{Q_z + G^{4z+2}}G}{\sqrt{2}(2z+1)\sigma^{2z}} + \frac{\gamma 2^{4z} E^{4z+1}(Q_z + G^{4z+2})L_{\max}}{2(2z+1)^2\sigma^{4z}}$$

$$+ \frac{4\eta_z^2\gamma\sigma^2 \sum_{j=1}^{d} L_j}{En}.$$
(35)

*Proof of Lemma 4.* First, we know from function smoothness that

$$f(x_t) - f(\bar{x}_t) \leq \langle \nabla f(\bar{x}_t), x_t - \bar{x}_t \rangle + \frac{\sum_{j=1}^{d} L_j(x_t(j) - \bar{x}_t(j))^2}{2}.$$
(36)

As can be seen from (36), we need to study the $x_t - \bar{x}_t$ in order to obtaining the upper bound for $f(x_t) - f(\bar{x}_t)$. Note that

$$x_t - \bar{x}_t = \frac{\gamma}{n} \sum_{i=1}^{n} \left(\eta_z\sigma\text{Sign}\left(\sum_{s=1}^{E} g_{t,s}^i + \sigma\xi_z\right) - \sum_{s=1}^{E} g_{t,s}^i\right).$$
(37)

For ease of presentation, we define that

$$\mathcal{A}_t^i \stackrel{\text{def.}}{=} \eta_z\sigma\text{Sign}\left(\sum_{s=1}^{E} g_{t,s}^i + \sigma\xi_z\right).$$
(38)

By taking the expectation over the random vector $\xi_z$, for any $j = 1, ..., d$, we have

$$\mathbb{E}_{\xi_z}[(x_t(j) - \bar{x}_t(j))^2] = \frac{\gamma^2}{n^2} \mathbb{E}_{\xi_z} \left[ \left( \sum_{i=1}^{n} \left( \mathcal{A}_t^i - \sum_{s=1}^{E} g_{t,s}^i(j) \right) \right)^2 \right] \tag{39a}$$

$$= \frac{\gamma^2}{n^2} \mathbb{E}_{\xi_z} \left[ \left( \sum_{i=1}^{n} \left( \mathcal{A}_t^i(j) - \mathbb{E}_{\xi_z} \left[ \mathcal{A}_t^i(j) \right] + \mathbb{E}_{\xi_z} \left[ \mathcal{A}_t^i(j) \right] - \sum_{s=1}^{E} g_{t,s}^i(j) \right) \right)^2 \right] \tag{39b}$$

$$\leq \frac{\gamma^2}{n^2} \mathbb{E}_{\xi_z} \left[ \left( \sum_{i=1}^{n} \left( \mathcal{A}_t^i(j) - \mathbb{E}_{\xi_z} \left[ \mathcal{A}_t^i(j) \right] \right) \right)^2 \right] \tag{39c}$$

$$+ \frac{\gamma^2}{n^2} \mathbb{E}_{\xi_z} \left[ \left( \sum_{i=1}^{n} \left( \mathbb{E}_{\xi_z} \left[ \mathcal{A}_t^i(j) \right] - \sum_{s=1}^{E} g_{t,s}^i(j) \right) \right)^2 \right], \tag{39d}$$

where the last inequality is obtained because $\sum_{i=1}^{n} \left( \mathcal{A}_t^i(j) - \mathbb{E}_{\xi_z} \left[ \mathcal{A}_t^i(j) \right] \right)$ is zero-mean and independent of $\sum_{i=1}^{n} \left( \mathbb{E}_{\xi_z} \left[ \mathcal{A}_t^i(j) \right] - \sum_{s=1}^{E} g_{t,s}^i(j) \right)$.

From (38) it is easy to check that $|\mathcal{A}_t^n(j)| \leq \eta_z^2 \sigma^2$. Hence, for the RHS of (39c), we have

$$\mathbb{E}_{\xi_z} \left[ \left( \sum_{i=1}^{n} \left( \mathcal{A}_t^i(j) - \mathbb{E}_{\xi_z} \left[ \mathcal{A}_t^i(j) \right] \right) \right)^2 \right] \overset{(a)}{=} \sum_{i=1}^{n} \mathbb{E}_{\xi_z} \left[ \left( \mathcal{A}_t^i(j) - \mathbb{E}_{\xi_z} \left[ \mathcal{A}_t^i(j) \right] \right)^2 \right]$$

$$\leq 2 \sum_{i=1}^{n} \left( \mathbb{E}_{\xi_z} \left[ \left( \mathcal{A}_t^i(j) \right)^2 \right] + \left( \mathbb{E}_{\xi_z} \left[ \mathcal{A}_t^i(j) \right] \right)^2 \right)$$

$$\leq 4 n \eta_z^2 \sigma^2, \tag{40}$$

where equality (a) is true because $\mathcal{A}_t^1(j), ..., \mathcal{A}_t^n(j)$ are independent to each other.

Therefore, from (39) and (40) we have

$$\mathbb{E}_{\xi_z} \left[ \sum_{j=1}^{d} L_j \left( x_t(j) - \bar{x}_t(j) \right)^2 \right] = \sum_{j=1}^{d} L_j \mathbb{E}_{\xi_z} \left[ \left( x_t(j) - \bar{x}_t(j) \right)^2 \right] \tag{41a}$$

$$\leq \frac{4 \eta_z^2 \gamma^2 \sigma^2 \sum_{j=1}^{d} L_j}{n}$$

$$+ \frac{\gamma^2}{n^2} \sum_{j=1}^{d} L_j \mathbb{E}_{\xi_z} \left[ \left( \sum_{i=1}^{n} \left( \mathbb{E}_{\xi_z} \left[ \mathcal{A}_t^i(j) \right] - \sum_{s=1}^{E} g_{t,s}^i(j) \right) \right)^2 \right] \tag{41b}$$

$$\leq \frac{4 \eta_z^2 \gamma^2 \sigma^2 \sum_{j=1}^{d} L_j}{n}$$

$$+ \frac{\gamma^2 L_{\max}}{n^2} \mathbb{E}_{\xi_z} \left[ \left\| \sum_{i=1}^{n} \left( \mathbb{E}_{\xi_z} \left[ \mathcal{A}_t^i \right] - \sum_{s=1}^{E} g_{t,s}^i \right) \right\|^2 \right]. \tag{41c}$$

To bound the RHS of (41c), we have

$$\mathbb{E}_{\xi_z}\left[\left\|\sum_{i=1}^n\left(\mathbb{E}_{\xi_z}\left[\mathcal{A}_t^i\right]-\sum_{s=1}^E g_{t,s}^i\right)\right\|^2\right]\leq n\sum_{i=1}^n\mathbb{E}_{\xi_z}\left[\left\|\mathbb{E}_{\xi_z}\left[\mathcal{A}_t^i\right]-\sum_{s=1}^E g_{t,s}^i\right\|^2\right]$$

$$\leq\frac{n}{4(2z+1)^2\sigma^{4z}}\sum_{i=1}^n\left\|\sum_{s=1}^E g_{t,s}^i\right\|_{4z+2}^{4z+2}, \qquad (42)$$

where the last inequality is due to Lemma 1.

Now we need to bound

$$\mathbb{E}\left[\left\|\sum_{s=1}^E g_{t,s}^i\right\|_{4z+2}^{4z+2}\right],$$

where the expectation is taken over both $\xi_z$ and the mini-batch gradient noise. To this end, we need the following lemma about the $\ell_p$-norm.

**Lemma 6.** *For any $M\in\mathbb{Z}_+$, $p>1$ and $M$ vectors $x_1,...,x_M\in\mathbb{R}^d$, we have*

$$\left\|\sum_{i=1}^M x_i\right\|_p^p\leq M^{p-1}\sum_{i=1}^M\|x_i\|_p^p. \qquad (43)$$

As a direct application of Lemma 6, we obtain

$$\mathbb{E}\left[\left\|\sum_{s=1}^E g_{t,s}^i\right\|_{4z+2}^{4z+2}\right]\leq\mathbb{E}\left[E^{4z+1}\sum_{s=1}^E\|g_{t,s}^i\|_{4z+2}^{4z+2}\right]=E^{4z+1}\sum_{s=1}^E\mathbb{E}\left[\|g_{t,s}^i\|_{4z+2}^{4z+2}\right] \qquad (44)$$

Then we can bound the RHS of (44) as

$$\mathbb{E}\left[\|g_{t,s}^i\|_{4z+2}^{4z+2}\right]=\mathbb{E}\left[\|g_{t,s}^i-\nabla f_i(x_{t,s-1}^i)+\nabla f_i(x_{t,s-1}^i)\|_{4z+2}^{4z+2}\right]$$

$$\overset{(a)}{\leq}\mathbb{E}\left[2^{4z+1}\|g_{t,s}^i-\nabla f_ti(x_{t,s-1}^i)\|_{4z+2}^{4z+2}+2^{4z+1}\|\nabla f_i(x_{t,s-1}^i)\|_{4z+2}^{4z+2}\right]$$

$$\overset{(b)}{\leq}2^{4z+1}Q_z+2^{4z+1}\|\nabla f_i(x_{t,s-1}^i)\|_2^{4z+2}$$

$$\overset{(c)}{\leq}2^{4z+1}(Q_z+G^{4z+2}), \qquad (45)$$

where inequality (a) follows Lemma 6, inequality (b) is due to Assumption 2, and inequality (c) is due to A.4 of Assumption 1.

Combing (41), (42), (44) and (45), we have

$$\mathbb{E}\left[\left\|\sum_{i=1}^n\left(\mathbb{E}_{\xi_z}\left[\mathcal{A}_t^i\right]-\sum_{s=1}^E g_{t,s}^i\right)\right\|\right]\leq\sqrt{\mathbb{E}\left[\left\|\sum_{i=1}^n\left(\mathbb{E}_{\xi_z}\left[\mathcal{A}_t^i\right]-\sum_{s=1}^E g_{t,s}^i\right)\right\|^2\right]}$$

$$\leq\sqrt{\frac{n^2 2^{4z}E^{4z+2}(Q_z+G^{4z+2})}{2(2z+1)^2\sigma^{4z}}}$$

$$\leq\frac{n2^{2z}E^{2z+1}\sqrt{(Q_z+G^{4z+2})}}{\sqrt{2}(2z+1)\sigma^{2z}} \qquad (46)$$

and

$$\mathbb{E}\left[\sum_{j=1}^d L_j\left(x_t(j)-\bar{x}_t(j)\right)^2\right]\leq\frac{4\eta_z^2\gamma^2\sigma^2\sum_{j=1}^d L_j}{n}+\frac{\gamma^2 L_{\max}}{n^2}\mathbb{E}\left[\left\|\sum_{i=1}^n\left(\mathbb{E}_{\xi_z}\left[\mathcal{A}_t^i\right]-\sum_{s=1}^E g_{t,s}^i\right)\right\|^2\right]$$

$$\leq\frac{4\eta_z^2\gamma^2\sigma^2\sum_{j=1}^d L_j}{n}+\frac{\gamma^2 2^{4z+1}E^{4z+2}(Q_z+G^{4z+2})L_{\max}}{4(2z+1)^2\sigma^{4z}}. \qquad (47)$$

Hence, we have

$$
\begin{aligned}
\mathbb{E}\left[f(x_t) - f(\bar{x}_t)\right] \leq & \mathbb{E}\left[\left\langle \nabla f(\bar{x}_t), \frac{\gamma}{n}\sum_{i=1}^{n}\left(\mathbb{E}_{\xi_z}\left[\mathcal{A}_t^i\right] - \sum_{s=1}^{E}g_{t,s}^i\right)\right\rangle\right] \\
& + \mathbb{E}\left[\frac{\sum_{j=1}^{d}L_j\left(x_t(j) - \bar{x}_t(j)\right)^2}{2}\right] \\
\leq & \|\nabla f(\bar{x}_t)\|\mathbb{E}\left[\left\|\frac{\gamma}{n}\sum_{i=1}^{n}\left(\mathbb{E}_{\xi_z}\left[\mathcal{A}_t^i\right] - \sum_{s=1}^{E}g_{t,s}^i\right)\right\|\right] \\
& + \mathbb{E}\left[\frac{\sum_{j=1}^{d}L_j\left(x_t(j) - \bar{x}_t(j)\right)^2}{2}\right] \\
\leq & \frac{\gamma 2^{2z}E^{2z+1}\sqrt{Q_z + G^{4z+2}}G}{\sqrt{2}(2z+1)\sigma^{2z}} + \frac{\gamma^2 2^{4z}E^{4z+2}(Q_z + G^{4z+2})L_{\max}}{4(2z+1)^2\sigma^{4z}} \\
& + \frac{2\eta_z^2\gamma^2\sigma^2\sum_{j=1}^{d}L_j}{n}.
\end{aligned}
\tag{48}
$$

$\square$

*Proof of Lemma 6.* To prove this lemma, we need to use a classical result on the monotonicity of $\ell_p$ norm:

**Lemma 7.** *(Kantorovich & Akilov, 2016) For any $x \in \mathbb{R}^d$ and $1 < r < p$, we have*

$$
\|x\|_p \leq \|x\|_r \leq d^{\frac{1}{r}-\frac{1}{p}}\|x\|_p.
\tag{49}
$$

Now from the definition of $\ell_p$ norm we have

$$
\begin{aligned}
\left\|\sum_{i=1}^{M}x_i\right\|_p^p &= \sum_{j=1}^{d}\left(\sum_{i=1}^{M}x_i(j)\right)^p \leq \sum_{j=1}^{d}\left(\sum_{i=1}^{M}|x_i(j)|\right)^p \\
&= \sum_{j=1}^{d}\|[x_1(j),...,x_M(j)]^\top\|_1^p \\
&\overset{(a)}{\leq} M^{p-1}\sum_{j=1}^{d}\|[x_1(j),...,x_M(j)]^\top\|_p^p \\
&= M^{p-1}\sum_{j=1}^{d}\sum_{i=1}^{M}(x_i(j))^p \\
&= M^{p-1}\sum_{i=1}^{M}\|x_i\|_p^p,
\end{aligned}
\tag{50}
$$

where inequality (a) is due to Lemma 7. $\square$

*Proof of Lemma 5.* First we unroll the difference $f(\bar{x}_t) - f(x_{t-1})$ into a telescopic sum across $E$ local steps.

$$f(\bar{x}_t) - f(x_{t-1}) = f(\bar{x}_{t-1,E}) - f(\bar{x}_{t-1,0}) = \sum_{s=1}^{E} f(\bar{x}_{t-1,s}) - f(\bar{x}_{t-1,s-1}) \tag{51a}$$

$$\leq \sum_{s=1}^{E} \left( -\langle \nabla f(\bar{x}_{t-1,s-1}), \bar{x}_{t-1,s-1} - \bar{x}_{t-1,s} \rangle + \frac{L_{\max}}{2} \|\bar{x}_{t-1,s} - \bar{x}_{t-1,s-1}\|^2 \right) \tag{51b}$$

$$= \sum_{s=1}^{E} \left( -\gamma \langle \nabla f(\bar{x}_{t-1,s-1}), \frac{1}{n} \sum_{i=1}^{n} g_{t-1,s}^i \rangle + \frac{\gamma^2 L_{\max}}{2} \left\| \frac{1}{n} \sum_{i=1}^{n} g_{t-1,s}^i \right\|^2 \right), \tag{51c}$$

where the inequality is due to the smoothness assumption. Taking expectation over the mini-batch gradient noise $g_{t-1,s}^1, ..., g_{t-1,s}^n$, for the first terms in (51c), we obtain

$$\mathbb{E} \left[ -\left\langle \nabla f(\bar{x}_{t-1,s-1}), \frac{1}{n} \sum_{i=1}^{n} g_{t-1,s}^i \right\rangle \right] = -\left\langle \nabla f(\bar{x}_{t-1,s-1}), \frac{1}{n} \sum_{i=1}^{n} \nabla f_i(x_{t-1,s-1}^i) \right\rangle \tag{52a}$$

$$= -\frac{1}{2} \|\nabla f(\bar{x}_{t-1,s-1})\|^2 - \frac{1}{2} \left\| \frac{1}{n} \sum_{i=1}^{n} \nabla f_i(x_{t-1,s-1}^i) \right\|^2 \tag{52b}$$

$$+ \frac{1}{2} \left\| \nabla f(\bar{x}_{t-1,s-1}) - \frac{1}{n} \sum_{i=1}^{n} \nabla f_i(x_{t-1,s-1}^i) \right\|^2. \tag{52c}$$

For the second terms in (51c), we have

$$\mathbb{E} \left[ \left\| \frac{1}{n} \sum_{i=1}^{n} g_{t-1,s}^i \right\|^2 \right] = \mathbb{E} \left[ \left\| \frac{1}{n} \sum_{i=1}^{n} g_{t-1,s}^i - \frac{1}{n} \sum_{i=1}^{n} \nabla f_i(x_{t-1,s-1}^i) + \frac{1}{n} \sum_{i=1}^{n} \nabla f_i(x_{t-1,s-1}^i) \right\|^2 \right]$$

$$\overset{(a)}{=} \mathbb{E} \left[ \left\| \frac{1}{n} \sum_{i=1}^{n} g_{t-1,s}^i - \frac{1}{n} \sum_{i=1}^{n} \nabla f_i(x_{t-1,s-1}^i) \right\|^2 \right] + \left\| \frac{1}{n} \sum_{i=1}^{n} \nabla f_i(x_{t-1,s-1}^i) \right\|^2$$

$$\overset{(b)}{=} \frac{1}{n^2} \sum_{i=1}^{n} \mathbb{E} \left[ \left\| g_{t-1,s}^i - \frac{1}{n} \sum_{i=1}^{n} \nabla f_i(x_{t-1,s-1}^i) \right\|^2 \right] + \left\| \frac{1}{n} \sum_{i=1}^{n} \nabla f_i(x_{t-1,s-1}^i) \right\|^2$$

$$\overset{(c)}{\leq} \frac{\zeta^2}{n} + \left\| \frac{1}{n} \sum_{i=1}^{n} \nabla f_i(x_{t-1,s-1}^i) \right\|^2, \tag{53}$$

where equalities (a) and (b) are true because the mini-batch gradient noise is independent, and inequality (c) is due to A.1 of Assumption 1.

Notice that owing to the function smoothness, we have for arbitrary $x, y \in \mathbb{R}^d$,

$$f(y) \leq \langle \nabla f(x), y - x \rangle + \frac{L_{\max}}{2} \|y - x\|^2, \tag{54}$$

which is equivalent to

$$\|\nabla f(x) - \nabla f(y)\| \leq L_{\max} \|y - x\|. \tag{55}$$

Now to bound the term in (52c), for every $s$, we have

$$\left\| \nabla f(\bar{x}_{t-1,s-1}) - \frac{1}{n} \sum_{i=1}^{n} \nabla f_i(x_{t-1,s-1}^i) \right\|^2$$

$$= \left\| \frac{1}{n} \sum_{i=1}^{n} \nabla f_i(\bar{x}_{t-1,s-1}) - \frac{1}{n} \sum_{i=1}^{n} \nabla f_i(x_{t-1,s-1}^i) \right\|^2$$

$$\leq \frac{L^2}{n} \sum_{i=1}^{n} \|\bar{x}_{t-1,s-1} - x_{t-1,s-1}^i\|^2$$

$$= \frac{\gamma^2 L_{\max}^2}{n} \sum_{i=1}^{n} \left\| \sum_{q=1}^{s-1} \left( \frac{1}{n} \sum_{j=1}^{n} g_{t-1,q}^j - g_{t-1,q}^i \right) \right\|^2$$

$$\leq \frac{(s-1)\gamma^2 L_{\max}^2}{n} \sum_{i=1}^{n} \sum_{q=1}^{s-1} \left\| \frac{1}{n} \sum_{j=1}^{n} g_{t-1,q}^j - g_{t-1,q}^i \right\|^2$$

$$\leq \frac{2(s-1)\gamma^2 L_{\max}^2}{n} \sum_{i=1}^{n} \sum_{q=1}^{s-1} \left( \left\| \frac{1}{n} \sum_{j=1}^{n} g_{t-1,q}^j \right\|^2 + \|g_{t-1,q}^i\|^2 \right)$$

$$\leq \frac{2(s-1)\gamma^2 L_{\max}^2}{n} \sum_{i=1}^{n} \sum_{q=1}^{s-1} \left( \frac{1}{n} \sum_{j=1}^{n} \|g_{t-1,q}^j\|^2 + \|g_{t-1,q}^i\|^2 \right). \tag{56}$$

For any $t = 1, ..., T$, $i = 1, ..., n$ and $q = 1, ..., s-1$, taking expectation over mini-batch gradient noise, we have

$$\mathbb{E}\left[\left\|g_{t-1,q}^j\right\|^2\right] = \mathbb{E}\left[\left\|g_{t-1,q}^i - \nabla f_i(x_{t-1,q-1}^i) + \nabla f_i(x_{t-1,q-1}^i)\right\|^2\right]$$

$$\leq \mathbb{E}\left[\left\|g_{t-1,q}^i - \nabla f_i(x_{t-1,q-1}^i)\right\|^2\right] + \left\|\nabla f_i(x_{t-1,q-1}^i)\right\|^2$$

$$\leq \zeta^2 + G^2. \tag{57}$$

Substituting (57) into (56), we have

$$\left\| \nabla f(\bar{x}_{t-1,s-1}) - \frac{1}{n} \sum_{i=1}^{n} \nabla f_i(x_{t-1,s-1}^i) \right\|^2 \leq 4(s-1)^2 \gamma^2 L_{\max}^2 (\zeta^2 + G^2). \tag{58}$$

Further substituting (53), (52) and (58) into (51) and by rearranging the terms, we obtain

$$\mathbb{E}[f(\bar{x}_t) - f(x_{t-1})] \leq \sum_{s=1}^{E} \left( -\frac{\gamma}{2} \|\nabla f(\bar{x}_{t-1,s-1})\|^2 + \frac{\gamma^2 L_{\max} - \gamma}{2} \left\| \frac{1}{n} \sum_{i=1}^{n} \nabla f_i(x_{t-1,s-1}^i) \right\|^2 \right)$$

$$+ \sum_{s=1}^{E} \left( \frac{\gamma^2 \zeta^2 L_{\max}}{2n} + \frac{\gamma}{2} \|\nabla f(\bar{x}_{t-1,s-1}) - \frac{1}{n} \sum_{i=1}^{n} \nabla f_i(x_{t-1,s-1}^i)\|^2 \right)$$

$$\overset{(a)}{\leq} -\frac{\gamma}{2} \sum_{s=1}^{E} \|\nabla f(\bar{x}_{t-1,s-1})\|^2 + \frac{E\gamma^2 \zeta^2 L_{\max}}{2n}$$

$$+ \sum_{s=1}^{E} 2(s-1)^2 \gamma^3 L_{\max}^2 (\zeta^2 + G^2), \tag{59}$$

where inequality (a) is by $(\gamma^2 L_{\max} - \gamma) \leq 0$.

Note that

$$\sum_{s=1}^{E}(s-1)^2 = \frac{(E-1)E(2E-1)}{6} \leq \frac{(E-1)E^2}{3}. \tag{60}$$

By applying it to (59), we finally have

$$\mathbb{E}[f(\bar{x}_t) - f(x_{t-1})] \leq -\frac{\gamma}{2}\sum_{s=1}^{E}\|\nabla f(\bar{x}_{t-1,s-1})\|^2 + \frac{E\gamma^2\zeta^2 L_{\max}}{2n} + \frac{2\gamma^3(E-1)E^2 L_{\max}^2(\zeta^2+G^2)}{3}. \tag{61}$$

$\square$

### C.4 PROOF OF THEOREM 3

We need a lemma similar to Lemma 4.

**Lemma 8.** *Suppose that Assumption 1 and 3 hold. For the $t$-th ($1 \leq t \leq T$) communication round in Algorithm 1, if $\eta = \sigma$ and $z = +\infty$, and $\sigma > E(G + Q_\infty)$, then*

$$\mathbb{E}[f(x_t) - f(\bar{x}_t)] \leq \frac{2\gamma^2\sigma^2 \sum_{j=1}^{d} L_j}{n}. \tag{62}$$

Following the similar idea as in the proof of Theorem 1, we have

$$\mathbb{E}[f(x_t) - f(x_{t-1})] = \mathbb{E}[f(x_t) - f(\bar{x}_t)] + E[f(\bar{x}_t) - f(x_{t-1})]$$

$$\leq -\frac{\gamma}{2}\sum_{s=1}^{E}\|\nabla f(\bar{x}_{t-1,s-1})\|^2 + \frac{E\gamma^2\zeta^2 L_{\max}}{2n}$$

$$+ \frac{2\gamma^3(E-1)E^2 L_{\max}^2(\zeta^2+G^2)}{3} + \frac{2\gamma^2\sigma^2 \sum_{j=1}^{d} L_j}{n}. \tag{63}$$

Rearranging the terms, we have

$$\frac{1}{E}\sum_{s=1}^{E}\|\nabla f(\bar{x}_{t-1,s-1})\|^2 \leq \frac{2\mathbb{E}[f(x_{t-1}) - f(x_t)]}{E\gamma} + \frac{\gamma\zeta^2 L_{\max}}{n}$$

$$+ \frac{4\gamma^2(E-1)EL_{\max}^2(\zeta^2+G^2)}{3} + \frac{4\gamma\sigma^2 \sum_{j=1}^{d} L_j}{En}. \tag{64}$$

Form the telescopic sum, we obtain

$$\mathbb{E}\left[\frac{1}{TE}\sum_{t=1}^{T}\sum_{s=1}^{E}\|\nabla f(\bar{x}_{t-1,s-1})\|^2\right] \leq \frac{2\mathbb{E}[f(x_0) - f^*]}{TE\gamma} + \frac{\gamma\zeta^2 L_{\max}}{n}$$

$$+ \frac{4\gamma^2(E-1)EL_{\max}^2(\zeta^2+G^2)}{3} + \frac{4\gamma\sigma^2 \sum_{j=1}^{d} L_j}{En}. \tag{65}$$

Here we provide a simple example to show that when $\sigma < E(G + Q_\infty)$, the algorithm cannot converge. Consider $E = 1$, $Q_\infty = 0$ and the problem

$$\min_{x\in\mathbb{R}}(x - A)^2 + (x + A)^2,$$

where $A > 0$ is some positive number. If we choose the initial to be $x_0 = \frac{A}{2}$. As one can see, the gradient at $x_0$ for the two parts of the objective function are $-A$ and $3A$, respectively. We denote that $\xi_\infty$ as the random noise following uniform distribution at $[-1, 1]$. If now $\sigma < A$, we have

$$\text{Sign}(-A + \sigma\xi_\infty) + \text{Sign}(3A + \sigma\xi_\infty) = 0, \tag{66}$$

i.e., this algorithm never update the variable.

*Proof of Lemma 8.* We first note that, when $z = +\infty$, we have

$$\Psi_\infty(x) = \begin{cases} x & x \in [-1, 1], \\ -1 & x < -1, \\ 1 & x > 1. \end{cases} \tag{67}$$

Again, from the smoothness assumption (A.2 in Assumption 1) we have,

$$f(x_t) - f(\bar{x}_t) \le \langle \nabla f(\bar{x}_t), x_t - \bar{x}_t \rangle + \frac{\sum_{j=1}^d L_j \left( x_t(j) - \bar{x}_t(j) \right)^2}{2}. \tag{68}$$

Taking expectation over $\xi_\infty$,

$$
\begin{aligned}
\mathbb{E}_{\xi_\infty}[x_t - \bar{x}_t] &= \mathbb{E}_{\xi_\infty} \left[ \frac{\gamma}{n} \sum_{i=1}^n \left( \sigma \mathrm{Sign} \left( \sum_{s=1}^E g_{t,s}^i + \sigma \xi_\infty \right) - \sum_{s=1}^E g_{t,s}^i \right) \right] \\
&\overset{(a)}{=} \frac{\gamma}{n} \sum_{i=1}^n \left( \sigma \Psi_\infty \left( \frac{\sum_{s=1}^E g_{t,s}^i}{\sigma} \right) - \sum_{s=1}^E g_{t,s}^i \right) \\
&\overset{(b)}{=} \frac{\gamma}{n} \sum_{i=1}^n \left( \sum_{s=1}^E g_{t,s}^i - \sum_{s=1}^E g_{t,s}^i \right) = 0,
\end{aligned} \tag{69}
$$

where equality (a) is because for any $x \in \mathbb{R}^d$, $\mathbb{E}_{\xi_\infty}[\mathrm{Sign}(x + \sigma \xi_\infty)] = \Psi_\infty(x/\sigma)$, equality (b) is due to $\sigma > \| \sum_{s=1}^E g_{t,s}^i \|_\infty$ almost surely and the property of the function $\Psi_\infty(\cdot)$ in (67).

For ease of presentation, we define that

$$\mathcal{B}_t^i \overset{\mathrm{def.}}{=} \sigma \mathrm{Sign} \left( \sum_{s=1}^E g_{t,s}^i + \sigma \xi_\infty \right). \tag{70}$$

From (69) we have learned that $\mathbb{E}_{\xi_\infty}[\mathcal{B}_t^i] = \sum_{s=1}^E g_{t,s}^i$. Thus, for any $j = 1, ..., d$, we have

$$
\begin{aligned}
\mathbb{E}_{\xi_\infty}[(x_t(j) - \bar{x}_t(j))^2] &\le \frac{\gamma^2}{n^2} \mathbb{E}_{\xi_\infty} \left[ \left( \sum_{i=1}^n \left( \mathcal{B}_t^i(j) - \mathbb{E}_{\xi_\infty} \left[ \mathcal{B}_t^i(j) \right] \right) \right)^2 \right] \\
&= \frac{\gamma^2}{n^2} \sum_{i=1}^n \mathbb{E}_{\xi_\infty} \left[ \left( \mathcal{B}_t^i(j) - \mathbb{E}_{\xi_\infty} \left[ \mathcal{B}_t^i(j) \right] \right)^2 \right] \\
&\le \frac{2\gamma^2}{n^2} \sum_{i=1}^n \left( \mathbb{E}_{\xi_\infty} \left[ \left( \mathcal{B}_t^i(j) \right)^2 \right] + \left( \mathbb{E}_{\xi_\infty} \left[ \mathcal{B}_t^i(j) \right] \right)^2 \right) \\
&\le \frac{4\gamma^2 \sigma^2}{n}.
\end{aligned} \tag{71}
$$

Finally, substituting (69) and (71) into (68), and taking the expectation over both $\xi_\infty$ and the mini-batch gradient noise, we have

$$
\begin{aligned}
\mathbb{E}[f(x_t) - f(\bar{x}_t)] &\le \mathbb{E}[\langle \nabla f(\bar{x}_t), x_t - \bar{x}_t \rangle] + \mathbb{E} \left[ \frac{\sum_{j=1}^d L_j \left( x_t(j) - \bar{x}_t(j) \right)^2}{2} \right] \\
&\le \frac{2\gamma^2 \sigma^2 \sum_{j=1}^d L_j}{n}.
\end{aligned} \tag{72}
$$

$\square$

# D EXPERIMENT DETAILS

## D.1 DETAILS FOR THE EXPERIMENT IN SECTION 4.2

In Table 3, we provide the tuned hyperparameters for all the tested algorithms on non-i.i.d MNIST. Specifically, we tuned the hyperparameters via grid search: $[0.1, 0.05, 0.01, 0.005]$ for stepsize, $[0, 0.3, 0.5, 0.7, 0.9]$ for the momentum coefficient, and $[0, 0.02, 0.05, 0.01, 0.03, 0.05, 0.1, 0.3, 0.5]$ for the noise scale.

| Algorithm | Stepsize | Momentum coefficient | Noise scale |
|---|---|---|---|
| SGDwM | 0.05 | 0.9 | |
| EF-SignSGDwM | 0.05 | 0.9 | |
| Sto-SignSGDwM | 0.01 | 0.9 | |
| SignSGD | 0.01 | 0 | 0 |
| 1-SignSGD | 0.01 | 0 | 0.05 |
| $\infty$-SignSGD | 0.01 | 0 | 0.05 |

Table 3: Hyperparameters used for FL on non-i.i.d MNIST.

In Figure 7, we visualize the performance of 1-SignSGD and $\infty$-SignSGD under different noise scales. As we can see, the results for 1-SignSGD and $\infty$-SignSGD are almost the same, except that the $\infty$-SignSGD is slightly better than 1-SignSGD when the noise scale is large.

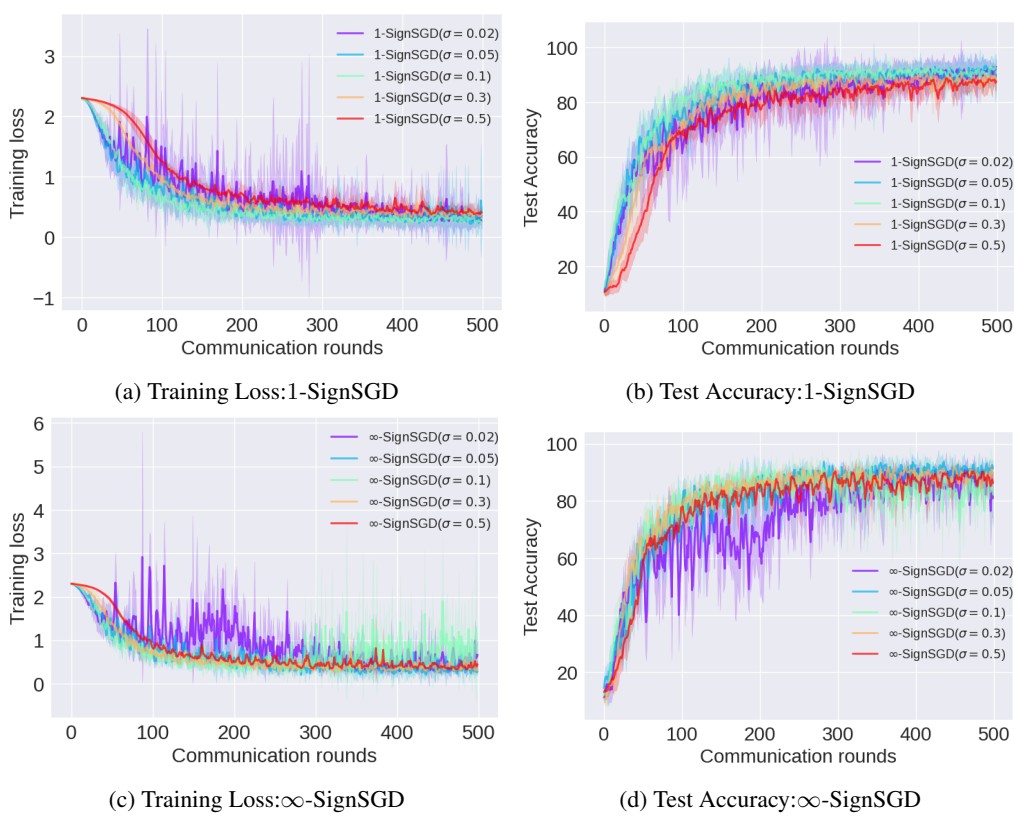

(a) Training Loss:1-SignSGD  (b) Test Accuracy:1-SignSGD

(c) Training Loss:$\infty$-SignSGD  (d) Test Accuracy:$\infty$-SignSGD

Figure 7: $z$-SignFedAvg under different noise scales on non-i.i.d MNIST

## D.2 DETAILS FOR THE EXPERIMENT IN SECTION 4.3

We denote the noiseless case, i.e., Algorithm 1 with $\sigma = 0$ as SignFedAvg.

**EMNIST:** For the experiment on EMNIST, we fixed the client stepsize as 0.05. We tuned the server stepsize, noise scales via grid search: $[1, 0.5, 0.1, 0.05, 0.01, 0.005]$ for stepsize, $[0, 0.005, 0.02, 0.05, 0.01, 0.03, 0.05, 0.1, 0.2]$ for noise scale. The comparison between 1-SignFedAvg and $\infty$-SignFedAvg on EMNIST is shown in Figure 8. The used hyperparameter in the Figure 4 and 8 are summarized in Table 4. We also visualize the performance of 1-SignFedAvg and $\infty$-SignFedAvg under various noise scales and local steps in Figure 9 and Figure 10.

**CIFAR-10:** For the experiment on CIFAR-10, we fixed the client stepsize as 0.1. We tuned the server stepsize, noise scales via grid search: $[10^0, 10^{-0.5}, 10^{-1}, 10^{-1.5}, 10^{-2}, 10^{-2.5}, 10^{-3}]$ for the stepsize, $[0, 0.0001, 0.0005, 0.001, 0.005]$ for the noise scale. The comparison between 1-SignFedAvg and $\infty$-SignFedAvg on CIFAR-10 is displayed in Figure 11. The used hyperparameter in the Figure 5 and 11 are summarized in Table 5. We also visualize the performance of 1-SignFedAvg and $\infty$-SignFedAvg under various noise scales and different numbers of local steps in Figure 12 and Figure 13. An interesting phenomeNon-in Figure 12 amd Figure 13 is that the more local steps are, the less impact the additive noise has on the convergence performance.

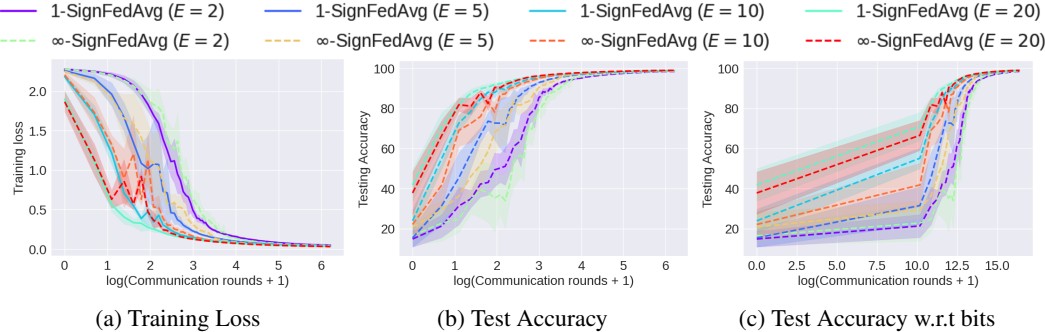

(a) Training Loss      (b) Test Accuracy      (c) Test Accuracy w.r.t bits

Figure 8: Performance of 1-SignFedAvg and $\infty$-SignFedAvg on EMNIST dataset.

| Algorithm | Server stepsize | Noise scale |
|---|---|---|
| 1-SignFedAvg | 0.03 | 0.01 |
| $\infty$-SignFedAvg | 0.03 | 0.01 |
| SignFedAvg | 0.03 | 0 |

Table 4: Hyperparameters for tested Algorithms on EMNIST.

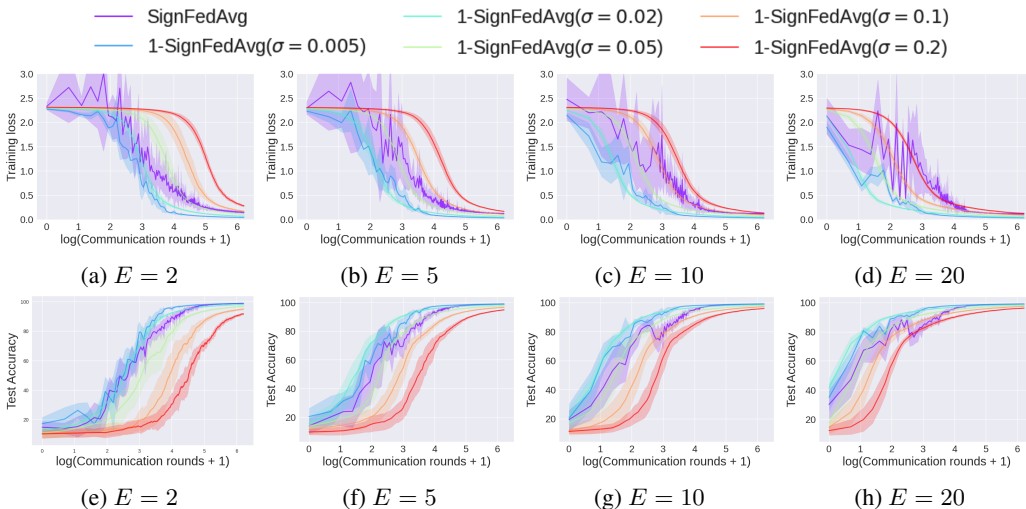

(a) $E = 2$    (b) $E = 5$    (c) $E = 10$    (d) $E = 20$

(e) $E = 2$    (f) $E = 5$    (g) $E = 10$    (h) $E = 20$

Figure 9: EMNIST: 1-SignFedAvg under different noise scales and different numbers of local steps

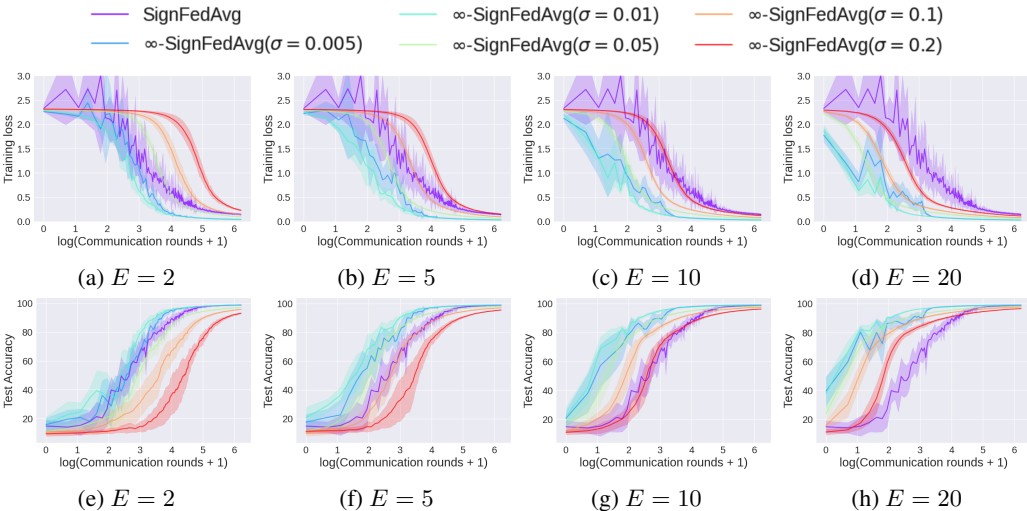

Figure 10: EMNIST: ∞-SignFedAvg under different noise scales and different numbers of local steps

| Algorithm | Server stepsize | Noise scale |
|---|---|---|
| 1-SignFedAvg | 0.0032 | 0.0005 |
| ∞-SignFedAvg | 0.0032 | 0.0005 |
| SignFedAvg | 0.0032 | 0 |

Table 5: Hyperparameters for tested Algorithms on CIFAR-10.

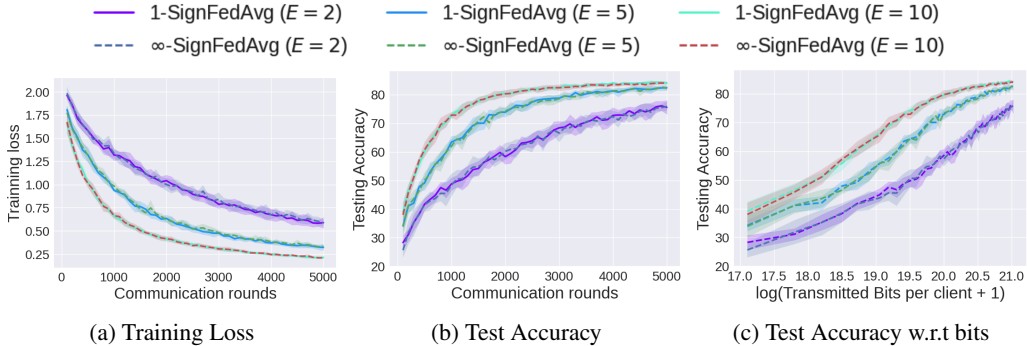

Figure 11: Performance of 1-SignFedAvg and ∞-SignFedAvg on CIFAR-10 dataset.

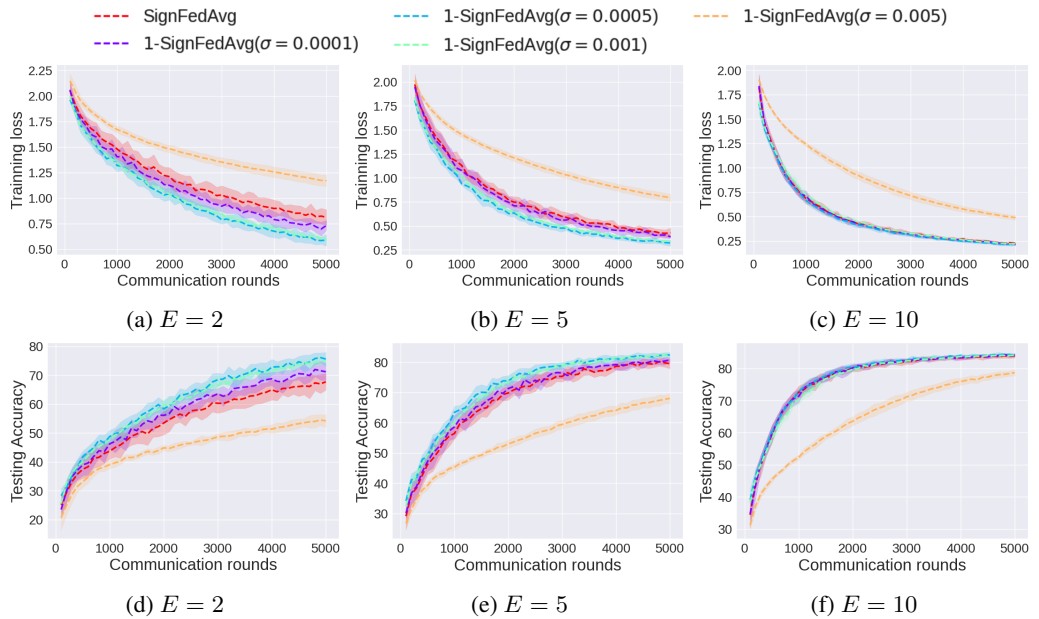

Figure 12: CIFAR-10: 1-SignFedAvg under different noise scales and different numbers of local steps

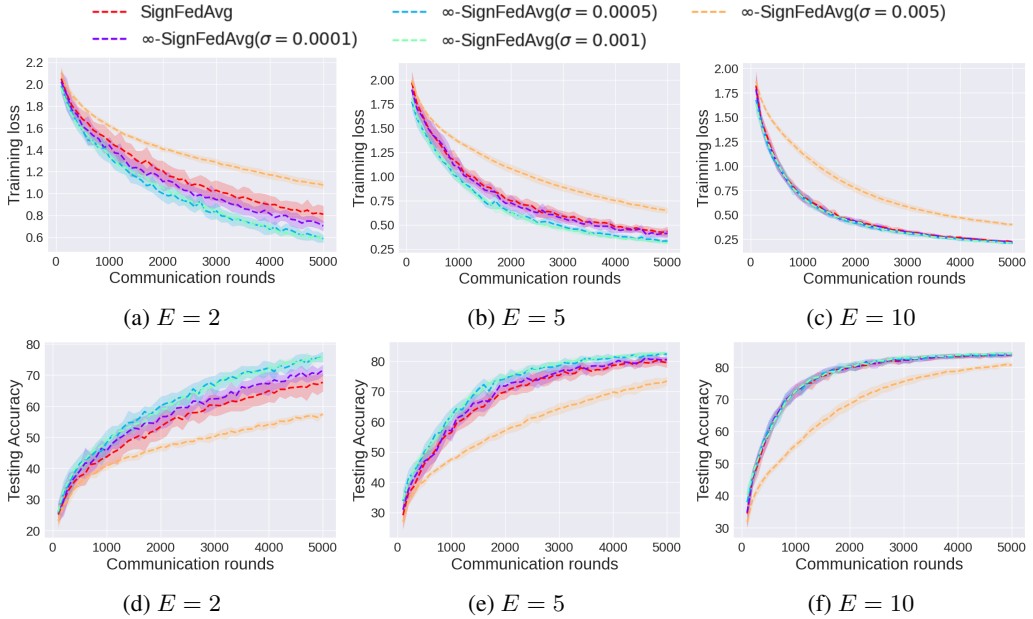

Figure 13: CIFAR-10: ∞-SignFedAvg under different noise scales and different numbers of local steps

### D.3 DETAILS FOR THE EXPERIMENT IN SECTION 4.4

For the experiment results shown in Figure 6, except for the noise scale, both 1-SignSGD/1-SignFedAvg and 1-SignSGD-plateau/1-SignFedAvg-plateau used the same hyperparameters found in previous experiments. In Table 6, we show the hyperparameters of the Plateau criterion for the adaptive noise scale, which are chosen by a few rounds of trial and error. Besides, we also show the corresponding test accuracy in Figure 14, and how the noise scale evolves over communication rounds in Figure 15.

| Dataset | $\sigma_{\text{init}}$ | $\sigma_{\text{bound}}$ | $\kappa$ | $\beta$ |
|---|---|---|---|---|
| Non-i.i.d. MNIST | 0.01 | 0.5 | 30 | 1.5 |
| EMNIST | 0.0001 | 0.1 | 10 | 2 |
| CIFAR-10 | 0.001 | 0.1 | 200 | 1.5 |

Table 6: Hyperparameters of Plateau criterion for three different datasets.

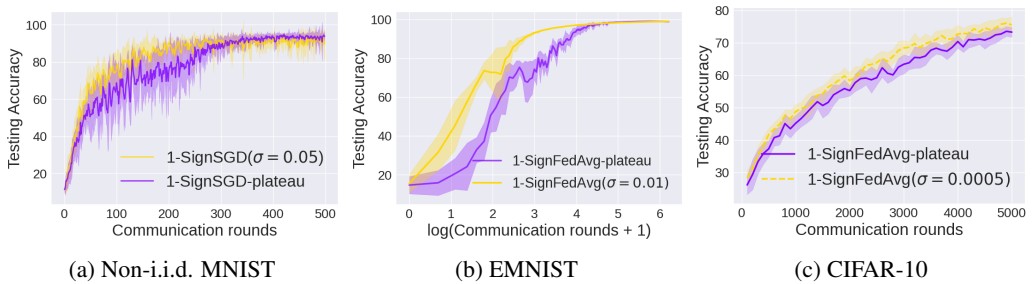

(a) Non-i.i.d. MNIST        (b) EMNIST        (c) CIFAR-10

Figure 14: The corresponding test accuracy to Figure 6.

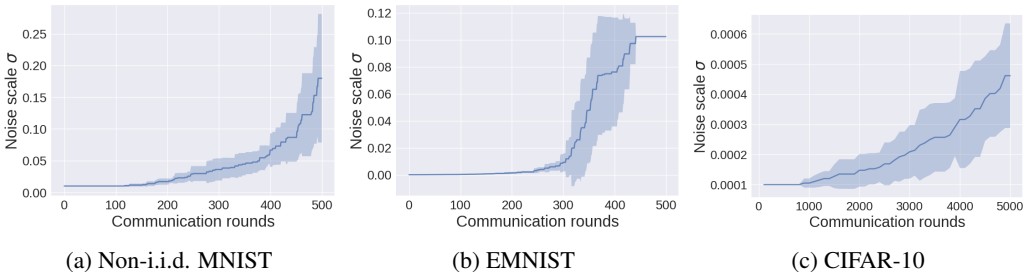

(a) Non-i.i.d. MNIST        (b) EMNIST        (c) CIFAR-10

Figure 15: The corresponding trends of noise scale to Figure 6.

## E COMPARISON WITH UNBIASED STOCHASTIC QUANTIZATION METHOD

In this part, we compare our Algorithm 1 to the QSGD (Alistarh et al., 2017) along with its extension to FedAvg, i.e., FedPAQ (Reisizadeh et al., 2020). As we have shown that $z$-SignSGD/$z$-SignFedAvg with the Gaussian noise and uniform noise behave very closely, here we only consider 1-SignSGD/1-SignFedAvg for comparison. We use the unbiased quantizer in (14) for both QSGD and FedPAQ.

We can see that the quantization level $s$ plays as a key role in the performance and communication efficiency of QSGD and FedPAQ. In a rough sense, $s$ also represents the number of bits needed to transmit for a single coordinate. Thus, we will compare our algorithms to them with different choices of $s$. We remark that, even in the most extreme case, i.e., $s = 1$, it still needs three alphabets $-1, 1, 0$ for communication, while sign-based method only uses $-1$ and $1$.

**Setting.** Again, we consider the three different datasets used in Section 4.2 and 4.2. Specifically, we compare the 1-SignSGD with QSGD on the non-i.i.d. MNIST dataset, and compare 1-SignFedAvg with FedPAQ on EMNIST and CIFAR-10. For all the algorithms, the client's stepsize and batchsize

are set to the same values used in Section 4.2 and 4.2. For 1-SignSGD/1-SignFedAvg, we reuse the previously found optimal hyperparameters. For QSGD, we tune the server stepsize via grid search on $[0.1, 0.05, 0.01, 0.005]$. For FedPAQ, we tune the server stepsize via grid search on $[1, 0.5, 0.1, 0.05, 0.01, 0.005]$. The chosen server stepsizes for QSGD and FedPAQ under three datesets are presented in Table 7.

| Algorithm | Non-i.i.d. MNIST | EMNIST | CIFAR-10 |
|---|---|---|---|
| QSGD($s = 1$) | 0.01 | | |
| QSGD($s = 2$) | 0.05 | | |
| QSGD($s = 4$) | 0.05 | | |
| FedPAQ($s = 1$) | | 1 | 1 |
| FedPAQ($s = 2$) | | 1 | 1 |
| FedPAQ($s = 4$) | | 1 | 1 |
| FedPAQ($s = 8$) | | 1 | 1 |

Table 7: The chosen server stepsizes for tested QSGD and FedPAQ on three datasets.

**Results.** From Figure 16, we can see that, our proposed sign-based compressor is consistently superior to the unbiased stochastic quantization method in low precision region (1 bit to 8 bits), except the only case that QSGD with $s = 4$ is slightly better than our 1-SignSGD on the non-i.i.d MNIST dataset. These results again, as (Bernstein et al., 2018; Karimireddy et al., 2019) did, show that the biased compressor, or more specifically the sign-based compressor, can be a strong competitor to those unbiased quantizer due to reduced variance. Our contribution in this work is to provide a generic framework that bridges the unbiased compressor and the biased one, which allows one to conveniently seek an optimal trade-off between the compression bias and variance.

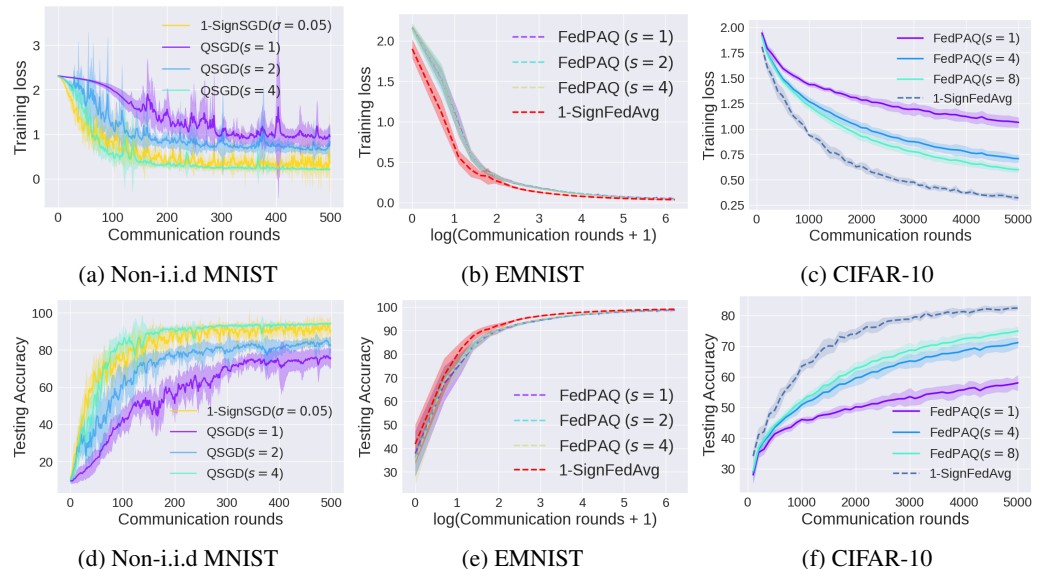

Figure 16: Comparison of 1-SignSGD/1-SignFedAvg with QSGD/FedPAQ on three datasets.

## F DIFFERENTIAL PRIVATE FEDERATED LEARNING ON EMNIST

Let us first review the definition of DP.

**Definition 3** (Approximate DP (Dwork et al., 2014))**.** *A randomized algorithm $M$ that takes as input a dataset consisting of individuals is $(\varepsilon, \delta)$-differentially private if for any pair of datasets $S, S'$ that differ in the record of a single individual, and for any event $E$,*

$$\mathbb{P}[M(S) \in E] \leqslant e^{\varepsilon} \mathbb{P}[M(S') \in E] + \delta. \tag{73}$$

The value $\varepsilon$ is regarded as the privacy budget, and the smaller it is the stronger privacy the algorithm provides. The quantity $\delta$ is usually set to $\frac{1}{n}$. The most popular mechanism to achieve DP is the Gaussian mechanism (Dwork et al., 2014). Specifically, similar to (Agarwal et al., 2021; Kairouz et al., 2021), here we consider client-level DP guarantee for Federated Learning, i.e, we regard each client as a single data point in Definition 3. Besides, we also adopt the local version of DP gurantee, i.e., each dataset in Definition 3 contains only one data point. Such DP guarantee do not assume that the server is trustworthy and hence is commonly used in practice (Agarwal et al., 2021; Kairouz et al., 2021). For more details on DP and its application in FL, we refer readers to (Dwork et al., 2014; Mironov, 2017; Abadi et al., 2016b; Geyer et al., 2017).

Here we describe the differential private version of Algorithm 1, which we term DP-SignFedAvg (Algorithm 2). The only difference between DP-SignFedAvg and $z$-SignFedAvg is that $z = 1$ is chosen (Gaussian noise), and the norm of local gradients is clipped before perturbing it by the noise and applying the sign compression. To obtain the client-level privacy guarantee, we adopt the privacy accounting method in (Mironov et al., 2019).

---

**Algorithm 2** DP-SignFedAvg

---

**Require:** Total communication rounds $T$, Number of local steps $E$, Number of clients $n$, Client sampling ratio $q$, Clients stepsize $\gamma$, Server stepsize $\eta$, Noise coefficient $\sigma$, Norm clipping coefficient $C$.
1: Initialize $x_0$ and for $i = 1, ..., n$.
2: **for** $t = 1$ to $T$ **do**
3:     Sample a set of clients $\mathcal{S}$ with size $qn$ for current round.
4:     **On Clients:**
5:     **for** $i$ in $\mathcal{S}$ **do**
6:         $x^i_{t-1,0} = x_{t-1}$
7:         **for** $s = 1$ to $E$ **do**
8:             $g^i_{t-1,s} = g_i(x^i_{t-1,s-1})$, where $g_i(\cdot)$ is the mini-batch gradient oracle of the $i$-th client.
9:             $x^i_{t-1,s} = x^i_{t-1,s-1} - \gamma g^i_{t-1,s}$.
10:         **end for**
11:         $\Delta^i_{t-1} = \text{Sign}\left(\frac{x_{t-1}-x^i_{t-1,E}}{\max\{1, \|x_{t-1}-x^i_{t-1,E}\|/C\}} + \mathcal{N}(0, \sigma^2 C^2 I)\right)$.
12:         Send $\Delta^i_{t-1}$ to the server.
13:     **end for**
14:     **On Server:**
15:     $x_t = x_{t-1} - \eta\frac{1}{n}\sum_{i=1}^{n}\Delta^i_{t-1}$.
16:     Broadcast $x_t$ to clients.
17: **end for**
18: **return** $x_T$.

---

Now we investigate the empirical performance of the DP-SignFedAvg on EMNIST, and compared it with the uncompressed DP-FedAvg used in (Agarwal et al., 2021; Kairouz et al., 2021).

**Settings.** We followed a setting similar to (Kairouz et al., 2021) for the experiment on EMNIST. We adopted the client-level differential privacy, i.e., to treat each client as a single data point, and perturbed the local gradients before sending them to server. We also used the technique of privacy amplification by client sub-sampling in (Kairouz et al., 2021; Geyer et al., 2017). For both DP-FedAvg and DP-SignFedAvg, the same CNN in Section 4.2 was used, and the maximum norm for clipping was set to 0.01. We sampled 100 clients at each communication round and ran both algorithms for 500 communication rounds. Similar to (Kairouz et al., 2021), we run the experiments under the privacy budgets $\varepsilon = [1, 2, 4, 6, 8, 10]$. In Table 8, we provide the hyperparameter for DP-FedAvg and DP-SignFedAvg for all levels of privacy budgets. Unlike previous experiments, the noise scales used in this experiment were determined by the privacy budget and the privacy accounting method in (Mironov et al., 2019).

**Results.** It can be seen from Figure 17 that DP-SignFedAvg is only slightly inferior to the uncompressed DP-FedAvg for various levels of privacy budget. It is worthy to note that the work (Kairouz et al., 2021) conducted a similar experiment and showed that the compressed DP-FedAvg

| Privacy budget | $\eta$ for DP-FedAvg | $\eta$ for DP-SignFedAvg | Noise scale |
|---|---|---|---|
| 1.0029 | 1 | 0.03 | 2.77 |
| 2.0171 | 2 | 0.05 | 1.57 |
| 4.0459 | 5 | 0.05 | 1.02 |
| 6.0135 | 5 | 0.05 | 0.845 |
| 8.0336 | 5 | 0.05 | 0.75 |
| 9.9996 | 5 | 0.05 | 0.685 |

Table 8: Hyperparameters for DP Algorithms on EMNIST.

with 12 bits for each gradient coordinate can be far worse than the uncompressed DP-FedAvg. It is a strong contrast to our DP-SignFedAvg which uses only 1 bit for each coordinate.

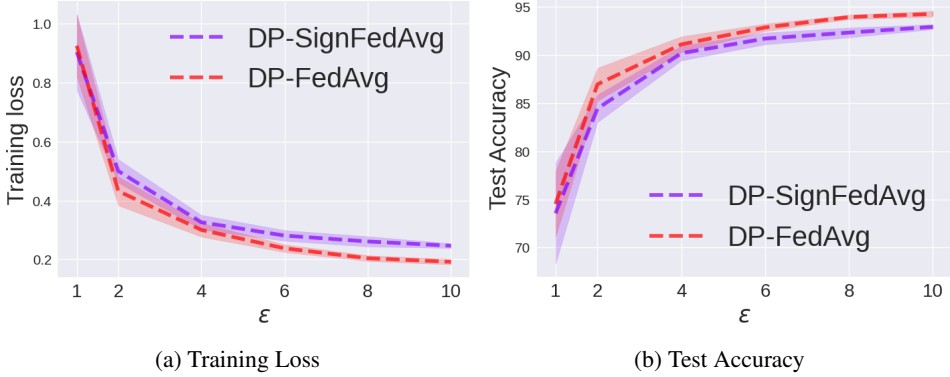

(a) Training Loss

(b) Test Accuracy

Figure 17: Performance of DP-SignFedAvg and DP-FedAvg

