# OpenReview forum: "$z$-SignFedAvg: A Unified  Stochastic Sign-based Compression for Federated Learning"
_ICLR.cc/2023/Conference — Submitted to ICLR 2023_

### Official Review · Reviewer_ExtX · 2022-10-20

**Confidence:** 3
**Correctness:** 4
**Technical Novelty And Significance:** 3
**Empirical Novelty And Significance:** 3
**Recommendation:** 6

**Clarity, Quality, Novelty And Reproducibility:**

The paper is well-written. The authors share sufficient details on the experimental setup and also the code.

**Strength And Weaknesses:**

Strengths:

- The authors study an important problem and their findings shed light on the success of other stochastic sign-based methods as well.
- The paper is theoretically sound, and the experimental results are consistent with most of the theoretical findings.

Weaknesses:
- By looking at Figure 2, it seems that the value of the noise scale should be very carefully adjusted to enjoy both a good convergence rate and a final performance. The paper misses a recipe on how to pick a good noise scale for a particular task with a particular architecture. The plots in Figure 2 also suggest that the performance could be improved with an adaptive strategy to update the noise scale. For instance, $\sigma=0.5$ has a faster convergence rate at the beginning of the training, but later it increases the loss, while $\sigma=2$ is slower in the beginning, but it reaches the best objective value at the end. So it would make sense to start with $\sigma=0.5$ and then gradually increase it to $\sigma=2$ for the best result. A recipe on how to pick the best noise scale and how to update it throughout training seems necessary to make the algorithm more practical. Perhaps the prior work [1] on how to match the noise and the compression level for the best utility of the data may help authors gain more intuition on this. It's probably hard to apply the idea in [1] (matching the entropy of the injected noise to the target distortion at the compression stage) to this problem directly since sign-based compression is not a "good" compressor in the rate-distortion theoretic sense. But it may still give an idea for a heuristic approach.

- It has been shown in many papers that the gradients tend to get sparser during the training, both in federated and centralized training. I believe this is another motivating factor to update the noise scale during training because the compression rate changes through iterations due to increasing sparsity. Again, please refer to [1] to see why the noise amount should be increased as the compression becomes more aggressive, or vice versa.

- It could be interesting to see how the proposed method performs compared to QSGD -- another famous stochastic quantization method.


[1] Isik, Berivan, and Tsachy Weissman. "Learning under Storage and Privacy Constraints." arXiv preprint arXiv:2202.02892 (2022).

**Summary Of The Paper:**

The authors propose a new Federated Averaging algorithm, called z-SignFedAvg, that compresses the model updates to their signs after perturbing them with noise. They show both theoretically and empirically that z-SignFedAvg enjoys a faster convergence rate than existing sign-based compression methods. The authors further exploit the noise injection mechanism and propose a modified algorithm that has differential privacy guarantees. They demonstrate their findings on EMNIST and CIFAR-10 datasets.

**Summary Of The Review:**

Overall, I enjoyed reading the paper and found the findings interesting. The proposed algorithm is also very promising, although it could be improved to pick the best noise scale at each iteration. I listed some of my ideas and suggestions above.

**Post rebuttal:** I think the new experiments on additional baselines improved the paper's empirical claims. The discussion on the noise scale also strengthened the paper. And I hope to see further investigation on this point in future work. After reading other reviews and authors' responses, I have decided to keep my original score 6.

---

> ### Author Response · Authors · 2022-11-18
> **Response to your concerns**
>
> **W1.**
>
> Comment 3.1:  By looking at Figure 2, it seems that the value of the noise scale should be very carefully adjusted to enjoy both a good convergence rate and a final performance. The paper misses a recipe on how to pick a good noise scale for a particular task with a particular architecture. The plots in Figure 2 also suggest that the performance could be improved with an adaptive strategy to update the noise scale. For instance, $\sigma$=0.5 has a faster convergence rate at the beginning of the training, but later it increases the loss, while $\sigma$=2 is slower in the beginning, but it reaches the best objective value at the end. So it would make sense to start with $\sigma$=0.5 and then gradually increase it to $\sigma$=2 for the best result. A recipe on how to pick the best noise scale and how to update it throughout training seems necessary to make the algorithm more practical. Perhaps the prior work [1] on how to match the noise and the compression level for the best utility of the data may help authors gain more intuition on this. It's probably hard to apply the idea in [1] (matching the entropy of the injected noise to the target distortion at the compression stage) to this problem directly since sign-based compression is not a "good" compressor in the rate-distortion theoretic sense. But it may still give an idea for a heuristic approach."
>
> Response: Thank you so much for your insightful suggestion! Inspired by your idea, we have proposed a heuristic strategy that can adapt the noise scale autonomously during the training process, which we termed the Plateau criterion. The idea is to start the algorithm with a relatively small noise scale and then increase it whenever the training loss stops improving for a few communication rounds. Please see the new Section 4.4 and the experiment results in Figure 6 in the revised manuscript.
>
>
> **W2.**
>
> Comment 3.2: It has been shown in many papers that the gradients tend to get sparser during the training, both in federated and centralized training. I believe this is another motivating factor to update the noise scale during training because the compression rate changes through iterations due to increasing sparsity. Again, please refer to [1] to see why the noise amount should be increased as the compression becomes more aggressive or vice versa.
>
>
> Response:  Thanks for brining this paper [1] to our attention! It is a truly useful reference for us because it justifies why the noise scale should be increased during the optimization process. We have included the discussion in the new Section 4.4 of the revised manuscript.
>
>
> **W3.**
>
> Comment 3.3: It could be interesting to see how the proposed method performs compared to QSGD -- another famous stochastic quantization method.
>
>
> Response:  Thank you for introducing this important method.
>
> In the revised manuscript, we have added a literature review for the three papers QSGD [2], FedPAQ [3] and FedCOM [4] in Section 1.1, and presented some details in Appendix A.
>
> In addition, we have conducted more experiments. Specifically, we have compared our proposed $z$-SignSGD/$z$-SignFedAvg with the QSGD on the Non-i.i.d MNIST dataset, and with the FedPAQ on the EMNIST and CIFAR-10.  Experiment details and the comparison results have been presented in Appendix E of the revised manuscript.
>
> From the new results in Figure 16, we can clearly see that our proposed sign-based methods are indeed superior to the unbiased compression methods, especially in the low precision region (use 1bit to 8bits for quantization per coordinate).
>
>
> [1] Isik, Berivan, and Tsachy Weissman. "Learning under Storage and Privacy Constraints." arXiv preprint arXiv:2202.02892 (2022).
>
> [2] Dan Alistarh, Demjan Grubic, Jerry Li, Ryota Tomioka, and Milan Vojnovic. Qsgd: Communication-
> efficient sgd via gradient quantization and encoding. Advances in neural information processing
> systems, 30, 2017.
>
> [3] Amirhossein Reisizadeh, Aryan Mokhtari, Hamed Hassani, Ali Jadbabaie, and Ramtin Pedarsani.
> Fedpaq: A communication-efficient federated learning method with periodic averaging and quanti-
> zation. In International Conference on Artificial Intelligence and Statistics, pp. 2021‚Äì2031. PMLR,
> 2020.
>
> [4] Farzin Haddadpour, Mohammad Mahdi Kamani, Aryan Mokhtari, and Mehrdad Mahdavi. Federated
> learning with compression: Unified analysis and sharp guarantees. In International Conference on
> Artificial Intelligence and Statistics, pp. 2350‚Äì2358. PMLR, 2021.

---

> ### Author Response · Authors · 2022-11-19
> **Have we resolved your concerns?**
>
> Dear Reviewer ExtX,
>
> We wonder whether our response addresses your current concerns? If not, please do let us know, and we are willing to write a further reply.

---

> > ### Comment · Reviewer_ExtX · 2022-12-11
> > **Thank you for the response.**
> >
> > I thank the authors for addressing most of my concerns and revising the manuscript accordingly. I think the new experiments and additional discussion on the noise scale improved the paper. After reading other reviewers' comments and the authors' responses, I have decided to keep my original score for acceptance.

---

### Official Review · Reviewer_M8Jc · 2022-10-25

**Confidence:** 4
**Clarity, Quality, Novelty And Reproducibility:** please see above.
**Correctness:** 3
**Technical Novelty And Significance:** 3
**Empirical Novelty And Significance:** 2
**Recommendation:** 3

**Strength And Weaknesses:**

S1. Stochastic 1-bit quantization, though being a popular compression strategy in many applilcation, has not been well investigated in FL. Thus, the paper provides a good application of such compressor to FL.

S2. The unified noise distribution is interesting and provides a general framework.

========================================================

W1. Adding Gaussian noise ($z=1$) leads to biased compressor, which in some sense loses the advantage of stochastic quantization. From the experiments we also see that it is alsmost always worse than uniform noise with $z=\infty$. This makes the practical implication of Gaussian noise limited.

W2. The theoretical results are not surprising and has been reported in literature. Basically, when the compression is biased ($z<\infty$), we have a bias term in the rate. If it is unbiased ($z=\infty$), then the rate can be $O(1/\sqrt{nTE})$. In the paper the authors stated "To the best of our knowledge, the previous works have never shown the sign-based method can achieve a linear-speedup convergence rate." Yet, the same convergence rate has been reported in literature, for example,

[1] Federated Learning with Compression:Unified Analysis and Sharp Guarantees, Haddadpour et al., AISTATS 2021.

Therefore, this claim is not true. In fact, the algorithm in this paper is also the same as that in the reference [1], except that in this paper stochastic sign is used instead of a general compressor. But since $z=\infty$ results in unbiased compression, the result in this paper is actually covered by the result in [1].

W3. In the experiments, I think the paper missed QSGD as an important and popular baseline method, since it is another way for unbiased compression. Also, how did you tune the learning rates? Could the slow convergence of sto-SignSGD in Figure 1 due to too large or small learning rate?

W4. Section 4.4 on DP variant seems unnecessary and incomplete. There is no formal analysis of the DP guarantee, nor the choice of noise variance, etc. I feel that this section may distract the main focus and contribution of the paper, plus this section itself is not well justified.

=====================================================

Questions:

Q1. The noise scale $\sigma$ in the paper seems important, since different $\sigma$ gives very different performances. How to properly tune this parameter in practice?

Q2. In Algorithm 1, line 10, why don't you multiply the Sign by $\eta_z\sigma$ as in (4)?

**Summary Of The Paper:**

The paper studies stochastic 1-bit (sign) quantization in federated learning. Particularly, the stochastic quantization is achieved by adding a random noise to the value and then taking the sign. The authors considers a general noise distreibution parameterized by $z$ with Gaussian ($z=1$) and uniform ($z=\infty$) as special cases. Convergence rates of applying this compressor to FL gradient uploading is provided. Experiments are conducted to show the effectiveness. The authors also discuss combining DP with the proposed algorithm at the end of the paper.

**Summary Of The Review:**

The paper considers an interesting problem of using stochastic unbiased sign in FL. Some new results are developed for the noise distribution in the quantizer. However, the theoretical analysis is existing in literature and same result has been reported before. The experimental evaluation misses an important unbiased quantization baseline, and the 'extra' section of DP approach seems incomplete. Thus, currently I'm biased to rejection.

---

> ### Author Response · Authors · 2022-11-18
> **Response to your concerns (1/2)**
>
> **W1**.
>
> Comment 2.1:  Adding Gaussian noise (z=1) leads to biased compressor, which in some sense loses the advantage of stochastic quantization. From the experiments we also see that it is almost always worse than uniform noise with $z=\infty$. This makes the practical implication of Gaussian noise limited.
>
> Response:
>
>  We thank you for the comment. Firstly, we are confused why you said that “From the experiments, we also see that it is almost always worse than uniform noise with $z=\infty$”. In fact, from our experiments, we observe that either Gaussian noise or the uniform noise the algorithms perform similarly, and sometimes one can be slightly better than the other (see Figure 1, which shows $z=\infty$ is better than $z=1$, and Figure 8, which shows $z=1$ is better $z=\infty$).
>
> Theoretically, it is also difficult to judge whether the uniform noise is better than the Gaussian noise. This is because while Theorem 2 suggests that the uniform case has a faster convergence rate than the Gaussian noise in Corollary 1, it requires a stronger Assumption 3 than Assumption 2 and the variance term could be much larger.
>
> Therefore, in our work, we don’t attempt to give a conclusion about which type of perturbation noise is better than the other. Instead, we prefer to highlight our contribution to proposing the unified noise perturbation scheme, which not only bridges the biased compression ($z<\infty$) and the unbiased compression ($z=\infty$) and enables us to develop the very first FedAvg-type algorithm with sign-based compression. As we mentioned previously, the existing compressed FedAvg algorithms mostly focus on unbiased compression. Thus, our work advances the state-of-the-art of compressed FL algorithms.
>
>
> **W2**.
>
> Comment 2.2:  The theoretical results are not surprising and has been reported in the literature. Basically, when the compression is biased ($z<\infty$), we have a bias term in the rate. If it is unbiased ($z=\infty$), then the rate can be O(1/nTE). In the paper the authors stated "To the best of our knowledge, the previous works have never shown the sign-based method can achieve a linear-speedup convergence rate." Yet, the same convergence rate has been reported in literature, for example,
>
> [1] Federated Learning with Compression: Unified Analysis and Sharp Guarantees, Haddadpour et al., AISTATS 2021.
>
> Response:
>
>  We thank you very much for bringing this paper to our attention. However, we believe that there is a misunderstanding about the connection between the paper [1] and our work.
>
> Paper [1] is indeed one of the representative papers about compressed FL algorithms. However, since paper [1] considers ``unbiased” compression, we did not consider them in the previous manuscript. Thanks to your comment, in retrospect, we should have discussed the distinctions between our work and [1] more and made proper comparisons.
>
> Firstly, paper [1] and other related works such as QSGD [2] and FedPAQ [3] all have assumed that the gradient compression is unbiased. Besides, they also have assumed that the compression error is bounded by the gradient norm (see, e.g., Assumption 2 of [1]). Unfortunately, this assumption do not hold for sign-based compression in general. ** Therefore, their convergence results are not applicable to the sign-based FL algorithms studied in our paper.**
>
> In fact, this is exactly the reason why there is still no FedAvg-type algorithm with sign-based compression proposed so far, even given the rich literature on sign-based methods, as mentioned in the introduction and related works sections.  The key breakthrough of our work is to introduce the unified noise perturbation scheme that overcomes the challenges and enables the development of the very first sign-based FedAvg algorithm.
>
>
> [2] Dan Alistarh, Demjan Grubic, Jerry Li, Ryota Tomioka, and Milan Vojnovic. Qsgd: Communication efficient sgd via gradient quantization and encoding. Advances in neural information processing systems, 30, 2017.
>
> [3] Amirhossein Reisizadeh, Aryan Mokhtari, Hamed Hassani, Ali Jadbabaie, and Ramtin Pedarsani. Fedpaq: A communication-efficient federated learning method with periodic averaging and quantization. In International Conference on Artificial Intelligence and Statistics, pp. 2021–2031. PMLR, 2020.

---

> > ### Author Response · Authors · 2022-11-18
> > **Response to your concerns (2/2)**
> >
> > **W3.**
> >
> > Comment 2.3:  In the experiments, I think the paper missed QSGD as an important and popular baseline method since it is another way for unbiased compression.
> >
> > Response:
> >
> > Thank you for introducing this important method. As we reply in the previous comment, we should not overlook these unbiased compression methods.
> >
> > In the revised manuscript, we have added a literature review for the three papers QSGD [2], FedPAQ [3] and FedCOM [4] in Section 1.1, and presented some details in Appendix A.
> >
> > In addition, we have conducted more experiments. Specifically, we have compared our proposed $z$-SignSGD/$z$-SignFedAvg with the QSGD on the Non-i.i.d MNIST dataset, and with the FedPAQ on the EMNIST and CIFAR-10.  Experiment details and the comparison results have been presented in Appendix E of the revised manuscript.
> >
> > From the new results in Figure 16, we can clearly see that our proposed sign-based methods are indeed superior to the unbiased compression methods, especially in the low precision region (use 1bit to 8bits for quantization per coordinate).
> >
> >
> > [4] Farzin Haddadpour, Mohammad Mahdi Kamani, Aryan Mokhtari, and Mehrdad Mahdavi. Federated learning with compression: Unified analysis and sharp guarantees. In International Conference on Artificial Intelligence and Statistics, pp. 2350‚Äì2358. PMLR, 2021.
> >
> >
> > Comment 2.4:  Also, how did you tune the learning rates?
> >
> > Response:
> >
> > As we mentioned in the first paragraph of Section 4.2, for each of the algorithms, we selected its best hyperparameters, including the stepsize, momentum coefficient and the noise scale, via grid search.
> >
> >
> > Comment 2.5:  Could the slow convergence of sto-SignSGD in Figure 1 due to too large or small learning rate?"
> >
> > Response:
> >
> > No, we don’t think so. In this experiment, we used the same stepsize for 1-SignSGD, $\infty$-SignSGD and Sto-SignSGD in Figure 1. As we mentioned in the paragraph right above Figure 1, it could be the large noise scale that slows the sto-SignSGD.
> >
> >
> >
> >
> > **W4.**
> >
> > Comment 2.6:  "Section 4.4 on DP variant seems unnecessary and incomplete. There is no formal analysis of the DP guarantee, nor the choice of noise variance, etc. I feel that this section may distract the main focus and contribution of the paper, plus this section itself is not well justified."
> >
> > Response:
> >
> > Thank you for the good suggestion. In view of your and Reviewer xh3m’s comments, we have decided not to emphasize too much on the DP version of SignFedAvg since it is merely a simple extension without significant theoretical improvement. Thus, we have moved all the contents about DP to Appendix F, leaving only a brief remark at the end of Section 3.
> >
> >
> > **Q1, Q2**.
> >
> > Comment 2.7:  The noise scale $\sigma$ in the paper seems important since different $\sigma$ give very different performances. How to properly tune this parameter in practice?
> >
> > Response:
> >
> > This is a very good point, and we thank you for raising it.
> >
> > Indeed, the noise scale plays an important role, and thus we suggest treating it as a tuning parameter like the stepsize. In the previous manuscript, we manually selected the noise scale by grid search. Through experiments, we find that a too large noise scale mostly slows down the convergence speed, while a too small noise scale can hurt the final performance.
> >
> > Motivated by this observation and inspired by your and Reviewer ExtX’s comments, we further present a heuristic Plateau criterion to adaptively find a proper noise scale. Please see the new Section 4.4 and the experiment results in Figure 6 in the revised manuscript.
> >
> > Comment 2.8:  In Algorithm 1, line 10, why don't you multiply the Sign by $\eta_z\sigma$ as in (4)?
> >
> > Response:
> >
> > That is a good point. Note that in Theorem 1, we set $\eta=\eta_z\sigma$. So, it is consistent with your intuition.

---

> ### Author Response · Authors · 2022-11-19
> **Have we resolved your concerns?**
>
> Dear Reviewer M8Jc,
>
> We wonder whether our response addresses your current concerns? If not, please do let us know, and we are willing to write a further reply.

---

### Official Review · Reviewer_xh3m · 2022-11-01

**Confidence:** 3
**Correctness:** 3
**Technical Novelty And Significance:** 2
**Empirical Novelty And Significance:** 2
**Recommendation:** 5

**Clarity, Quality, Novelty And Reproducibility:**

* The presentation is decent and understandable. I did not go through the detailed proofs to check the correctness, but there does not appear to be any discrepancy in results. On the other hand, the assumptions are quite strong, and the questions mentioned under ‘Weaknesses’ need to be discussed.

* The extension of SignSGD to SignFedAvg is a logical next step. The key novelty is in terms of analysis of $z$-distribution and convergence analysis of SignFedAvg.


**Strength And Weaknesses:**

**Strengths:**
1. Applying sign-based compression on top of model updates is a natural extension of signSGD [Bernstein et al. 2018], where compression is applied on gradients. This is conceptually simple and can be practically useful.

**Weaknesses:**
1. For the case of $z = +\infty$, i.e., when the injected noise is uniformly distributed on $[-1,+1]$, Assumption 3 seems very strong. It is also required for the theoretical result that the variance of the noise should be substantially large, i.e., $\sigma > E(G + Q_{\infty})$. This large variance in noise, in turn, results in a large variance term of the convergence result (Theorem 3 in Appendix B, formal version of Theorem 2). Remark 2 claims that SignFedAvg has the same order-wise convergence rate as that f uncompressed FedAvg. However, it is crucial that the large variance of Theorem 3 is discussed.

* More specifically, the variance term in Theorem 3 is $\frac{4\gamma\sigma^2\sum_{j=1}^{d}L_j}{En}$. The theorem requires that $\sigma > E(G + Q_{\infty})$, which implies that the variance term is at least $\frac{4\gamma G^2 (E + Q_{\infty})^2 \sum_{j=1}^{d}L_j}{En} \geq \frac{4\gamma G^2 E \sum_{j=1}^{d}L_j}{n}$.

    * When $L_j$’s are constant, the variance terms is dimension dependent. Since large neural network have significantly large $d$, the variance would be quite substantial.

   * It is not clear why the variance term behaves as $O((n\tau)^{-1/2})$ if $E \leq n^{-3/4}\tau^{1/4}$. It is important to give further details.

   * The authors compare the convergence result with [Yu et al. 2019], but the variance term in that paper is proportional to the variance of SGD operator. On the other hand, in Theorem 3, the variance term depends on $d$ and $Q_{\infty}$, which can be really large.

2. The noise variance $\sigma$ is considered as a hyperparameter, and for EMNIST and CIFAR values of $\sigma$ chosen via hyperparmater tuning are quite small (0.01 and 0.005). Theorem 3, on the other hand, shows counter-examples that the algorithm cannot converge for small values of $\sigma$. This seems to indicate a gap between theory and experiments. More discussion on this point would be really helpful.

3. In DP-SignFedAvg (Algorithm 2), the DP noise is added only on the local update. This will give client-level privacy and not item-level privacy. In other words, DP guarantee on the aggregate model at each round against adding or removing a client should. The paper does not mention any of these details, but only has Definition 2, which is generic and does not specify how to apply it to FL.


**Summary Of The Paper:**

The paper considers sign-based compression for federated learning (FL). It proposes a stochastic sign-based compressor, where first a random noise from (the proposed) $z$-distribution is added and then a sign operator is applied. The $z$-distribution covers Gaussian and uniform noise as its special cases. The sign-based compression is applied on top of the classical FedAvg, where clients use local SGD to compute model updates and the server computes a weighted average of updates. SignFedAvg can be viewed as an extension of SignSGD in [Bernstein et al. 2018] to FedAvg.

**Summary Of The Review:**

The analysis of sign-based compression with $z$-distribution noise is interesting. At the same time, the convergence analysis relies on very strong assumptions and leaves open several questions. Further, the DP part needs to be fleshed out. Therefore, the paper is not yet ready for publication.

---

> ### Author Response · Authors · 2022-11-18
> **Response to your concerns. (1/2)**
>
> **S1**
>
> Comment 1.0: Applying sign-based compression on top of model updates is a natural extension of signSGD [Bernstein et al. 2018], where compression is applied on gradients. This is conceptually simple and can be practically useful.
>
> Response:
>
>  We thank you for your detailed review and for providing constructive comments. Indeed, extending signSGD to our SignFedAvg is natural and conceptually simple. However, we should emphasize that this extension can be technically non-trivial.
>
> In fact, as we mentioned in the introduction and related works sections, there have been many efforts to improve the sign-based methods, but none of them have ever proposed the FedAvg-type algorithm for sign-based compression. Intriguingly, FedAvg algorithms with unbiased compression have already been proposed a few years ago; see [1], [2] and 3]. The main obstacle to having a sign-based FedAvg algorithm is that sign-based compression is unbiased in nature. The key breakthrough of our work is to introduce the unified noise perturbation scheme that explicitly characterizes how the noise scale affects the bias due to sign compression and therefore builds the connection between biased sign operation [4] and unbiased compression [1].
>
> Thus, our work not only presents a practically useful algorithm but also provides a novel unified point of view and a set of new convergence analyses for the sign-based FL algorithms. We hope that, with the explanation, our technical contribution in the manuscript can be better recognized.
>
> Thanks to your and other reviewers’ insightful comments, we have significantly revised the manuscript.
>
>
>
> **W1**
>
> Comment 1.1:  For the case of $z=+\infty$, i.e., when the injected noise is uniformly distributed on [-1,+1], Assumption 3 seems very strong. It is also required for the theoretical result that the variance of the noise should be substantially large.
>
> Response:
>
> Indeed, you are correct that our Assumption 3 is slightly stronger than the commonly used second-order moment condition on the minibatch gradient noise. Nevertheless, we think it is arguable that Assumption 3 is still reasonable in practice. This is because it rarely happens that the minibatch gradient noise lies in some unbounded region, and therefore it should be acceptable to assume that minibatch gradient noises are bounded with probability one.
>
> About the large noise scale, please see our response to Comments 1.2 below.
>
> Comment 1.2:  When $L_j$’s are constant, the variance terms is dimension dependent. Since large neural network have significantly large $d$, the variance would be quite substantial.
>
> Response:
>
> This is a very good point! In fact, such dependence on $\sum_{i=1}^d{L_i}$ is common in the convergence analysis for all the existing sign-based algorithms like SignSGD[4] and  Sto-SignSGD[5]. This should not be surprising because the binary quantization is applied to the gradient vector in an element-by-element fashion.
>
> While such dimension-dependent variance term is undesirable, it is found that the sign-based methods can still work well in practice for large-scale neural networks. This is not only observed in our work (see Figure 4,5 in Section 4) but also in other papers such as [4] and [5]. A potential reason is that the gradient tends to be sparse, and thereby, the ``effective dimension” of the quantization error is much less than $d$.
>
> Comment 1.3:  It is not clear why the variance term behaves as $O((n\tau)^{−1/2})$ if $E\leq n^{−3/4}\tau^{1/4}$. It is important to give further details.
>
> Response:
>
> Notice that the third term in the RHS of (18) is $O(\frac{E^2n}{\tau})$. Thus, if we let $E\leq n^{-3/4}\tau^{1/4}$, then it makes it have the order  $O(\frac{1}{(n\tau)^{\frac12}})$.
>
> We have revised the sentence below (18) to make this point clear.
>
> Comment 1.4:  The authors compare the convergence result with [Yu et al. 2019], but the variance term in that paper is proportional to the variance of SGD operator. On the other hand, in Theorem 3, the variance term depends on $d$ and $Q_\infty$, which can be really large.
>
> Response:
>
> This is exactly one of our main key breakthroughs. In the presence of the additional variance term caused by the sign-based compression, our work is one of the first to show that the sign-based method injected with enough uniform noise can achieve the same convergence rate order as the uncompressed FedAvg algorithm in [Yu et al. 2019].
>
> As we discussed after Remark 2, [Jin et al., 2020] and [Safaryan & Richtarik, 2021] also presented similar results, but their results are based on different convergence metrics from [Yu et al. 2019]. Our results are based on the same convergence metric as [Yu et al. 2019].
> Again, as we said in the response to Comment 1.2, comparing the uncompressed algorithm, we found in practice that the variance term affect little on the convergence performance.

---

> > ### Author Response · Authors · 2022-11-19
> > **Response to your concerns. (2/2)**
> >
> > **W2**.
> >
> > Comment 1.5:  The noise variance $\sigma$ is considered as a hyperparameter, and for EMNIST and CIFAR values of $\sigma$ chosen via hyperparmater tuning are quite small (0.01 and 0.005). Theorem 3, on the other hand, shows counter-examples that the algorithm cannot converge for small values of $\sigma$. This seems to indicate a gap between theory and experiments. More discussion on this point would be really helpful.
> >
> > Response:
> >
> >  This is a very good observation, and we thank you for raising it.
> >
> > Indeed, while it is well-known that the convergence analysis usually is for the worst case, and therefore there is always a gap between theorems and practice, we find that there could be another factor that allows us to use a small noise scale in the experiments.
> >
> > Specifically, we conjecture that the minibatch gradient noise itself may function as the perturbation noise for the sign operation in Eqn. (2). In fact, as shown in [Chen et al], the minibatch gradient noise approximately follows a symmetric distribution. Therefore, from Eqn.  (2) we can see that the minibatch gradient noise itself may also help to mitigate the bias in sign-based compression. Thus, the optimal noise scale in real applications can be much smaller than the one suggested by the theorems, as also confirmed in our experiments.
> >
> > Therefore, it is possible to refine the convergence analysis to explicitly consider the impact of the minibatch gradient noise on the noise scale. Since it is generally unrealistic to place any symmetric assumption on the distribution of minibatch gradient noise, and also due to a limited time, we would like to leave this as a future direction. In the manuscript, we have added a new Remark 4 in Section 3.2 to discuss the impact of the minibatch gradient noise on the noise scale.
> >
> >
> > **W3**.
> >
> > Comment 1.6:
> >
> > In DP-SignFedAvg (Algorithm 2), the DP noise is added only on the local update. This will give client-level privacy and not item-level privacy. In other words, DP guarantee on the aggregate model at each round against adding or removing a client should. The paper does not mention any of these details, but only has Definition 2, which is generic and does not specify how to apply it to FL.
> >
> > Response:
> >
> > Thank you for the comments. First, in view of your and Reviewer M8jc comments, we have decided not to emphasize too much on the DP version of SignFedAvg since it is merely a simple extension without significant theoretical improvement. Thus, we have moved all the contents about DP to Appendix F, leaving only a brief remark at the end of Section 3.
> >
> > Second, we said in the setting on page 9 of the original version that we use user-level DP, which is synonymous with client-level DP. We have provided a few more details in Appendix F on the experiment of DP-SignFedAvg, and also revised the term "user-level DP" to "client-level" DP in order to avoid confusion.
> >
> > [1] Dan Alistarh, Demjan Grubic, Jerry Li, Ryota Tomioka, and Milan Vojnovic. Qsgd: Communication- efficient sgd via gradient quantization and encoding. Advances in neural information processing systems, 30, 2017.
> >
> > [2] Amirhossein Reisizadeh, Aryan Mokhtari, Hamed Hassani, Ali Jadbabaie, and Ramtin Pedarsani. Fedpaq: A communication-efficient federated learning method with periodic averaging and quantization. In International Conference on Artificial Intelligence and Statistics, pp. 2021‚Äì2031. PMLR, 2020.
> >
> > [3] Farzin Haddadpour, Mohammad Mahdi Kamani, Aryan Mokhtari, and Mehrdad Mahdavi. Federated learning with compression: Unified analysis and sharp guarantees. in International Conference on Artificial Intelligence and Statistics, pp. 2350‚Äì2358. PMLR, 2021.
> >
> > [4] Jeremy Bernstein, Yu-Xiang Wang, Kamyar Azizzadenesheli, and Animashree Anandkumar. signsgd: Compressed optimisation for non-convex problems. In International Conference on Machine Learning, pp. 560‚Äì569. PMLR, 2018.
> >
> > [5] Mher Safaryan and Peter Richtarik. Stochastic sign descent methods: New algorithms and better theory. In International Conference on Machine Learning, pp. 9224‚Äì9234. PMLR, 2021.

---

> ### Author Response · Authors · 2022-11-19
> **Have we resolved your concerns?**
>
> Dear Reviewer xh3m,
>
> We wonder whether our response addresses your current concerns? If not, please do let us know, and we are willing to write a further reply.

---

> > ### Comment · Reviewer_xh3m · 2022-12-01
> > **Thank you**
> >
> > I thank the authors for answering my questions. For theoretical convergence results, both the convergence rate as well as variance terms are important. This is because variance term provides the error term, which governs the step size and the number of local/global iterations. Due to strong assumptions and a large variance term, I still believe that the theoretical results are somewhat limited. Also, all reviewers and the authors agree that DP results play a minor role in the paper. Having said this, the authors have addressed some of my concerns, and thus I am increasing my score.

---

> > > ### Author Response · Authors · 2022-12-02
> > > **Thank you for acknowledging our response! But we think the variance terms are not necessarily large.**
> > >
> > > Thank you for acknowledging our response and revision!
> > >
> > > Firstly, please allow us to discuss more the variance term in the convergence analysis. Firstly, such variance term commonly occurs in all existing distributed sign-based methods. Though not proven yet, we conjecture that such variance can never be removed due to the element-by-element sign operations. Moreover, as is also corroborated by the experiment results, we argue that the variance term might not dominate the algorithm convergence. Because the variance terms in our theoretical results are not necessarily large. For example,   for the case $z=+\infty$, from (19) we can see that the convergence bound of uncompressed algorithm has a term depending on $\zeta^2L_\max$, where $\zeta^2=\mathbb E[||g(x)-\nabla f(x)||^2]$. When the scale of gradient noise is constant (or approximately constant) at every coordinate, i.e, $ |g(x)_i-\nabla f(x)_i|\approx some\ constant $,  then $\zeta^2L_\max \approx Q_\infty^2 dL_\max$. Therefore, we can see that in this case, if $ dL_\max \gg \sum_1^d L_i$ ,  the additional variance term in (19) caused by the sign compression could be ignorable.
> > >
> > >
> > > We still want to take this opportunity to elaborate more on our novelty and technical contributions. Indeed, we agree that our presented convergence analysis requires a slightly stronger (but not unrealistic) assumption. Nonetheless, they should not dim the technical novelty and contributions of our work. In particular, we should emphasize that our work presents a novel noise perturbation scheme (Section 2) that not only unifies the existing stochastic sign-based methods but also reveals the bias-variance trade-off of the sign operator, both of which have never appeared in the literature.
> > >
> > > More importantly, our contributions are not only in theory, but also in practice. Such a new theoretically-driven scheme enables us to extend the sign-based compression to the FedAvg algorithm (i.e., the proposed z-signFedAvg algorithm) to improve communication efficiency, which is another breakthrough compared to existing stochastic sign-based methods. What is more exciting is to see that z-signFedAvg can perform almost as well as uncompressed FedAvg in practical experiments with a significant amount of bits saved.  Such results are not known until this work.
> > >
> > >
> > > Lastly, we believe that our work significantly advances the state-of-the-art of sign-based federated learning algorithms and hope that the technical novelty and contributions can be recognized.

---

### Author Response · Authors · 2022-11-18
**Summary for paper revision**

# Summary for paper revision

We sincerely thank the reviewers for their constructive and insightful comments. We have revised our manuscript thoroughly. The revisions of our manuscript are all marked with a blue color. Here let us briefly summarize the major revisions in the manuscript.

1. Relegating the DP part to Appendix: As raised by Reviewer xh3m and M8jc, the DP part in the original version plays a minor role and may distract from the flow of the main content. Therefore, we have moved all the contents about DP to Appendix F, leaving only a brief remark at the end of Section 3. Besides, as a response to Reviewer xh3m, we have provided a few more details in Appendix F on the experiment of DP-SignFedAvg, and also revised the term "user-level DP" to "client-level" DP in order to avoid confusion.

2. Comparison to existing unbiased quantization methods: As concerned by Reviewer M8jc and Reviewer ExtX, in the previous manuscript, we overlooked the popular unbiased stochastic quantization methods. Therefore, in the revised version, we have made the following modifications:

- Add a literature review for the three papers QSGD [1], FedPAQ [2], and FedCOM [3] in Section 1.1, and discuss the specific quantization schemes with details in Appendix A.

- Add experiment comparison results with the QSGD on the Non-i.i.d MNIST dataset and FedPAQ on the EMNIST and CIFAR-10 in Appendix E.

3. Providing a new adaptive strategy for tuning the noise scale: As concerned by Reviewer ExtX and Reviewer M8jc, in the previous manuscript, we did not show how to pick the noise scale efficiently. In Section 4.4, we have proposed a practically useful strategy to adapt the noise scale autonomously during the training process. We have also verified its efficacy by conducting more experiments on Non-i.i.d MNIST, EMNIST, and CIFAR-10 datasets. Experiment results have been added to Appendix D3.

4. More discussions on theoretical results in Section 3.4: This is mainly in response to the two concerns raised by Reviewer xh3m, and also to other readers who may share the same ones. Firstly, we have emphasized why Assumptions 2 and 3 made in the paper, though stronger than the one commonly used in the literature, are not unrealistic in practice. Then, more importantly, we have argued that the minibatch gradient noise itself can work as a perturbation noise for the sign-based compression and, therefore, somehow explains why the noise scale used in our experiments can be much smaller than the one suggested in the theorems.

Lastly, to prove that our newly added results are reproducible, we have also updated the codes in the supplementary material accordingly.

[1] Dan Alistarh, Demjan Grubic, Jerry Li, Ryota Tomioka, and Milan Vojnovic. Qsgd: Communication-
efficient sgd via gradient quantization and encoding. Advances in neural information processing
systems, 30, 2017.

[2] Amirhossein Reisizadeh, Aryan Mokhtari, Hamed Hassani, Ali Jadbabaie, and Ramtin Pedarsani.
Fedpaq: A communication-efficient federated learning method with periodic averaging and quanti-
zation. In International Conference on Artificial Intelligence and Statistics, pp. 2021–2031. PMLR,
2020.

[3] Farzin Haddadpour, Mohammad Mahdi Kamani, Aryan Mokhtari, and Mehrdad Mahdavi. Federated
learning with compression: Unified analysis and sharp guarantees. In International Conference on
Artificial Intelligence and Statistics, pp. 2350–2358. PMLR, 2021.

---

### Decision · Program_Chairs · 2023-01-20

**Decision:**

Reject

**Justification For Why Not Higher Score:**

see above

**Justification For Why Not Lower Score:**

N/A

**Metareview: Summary, Strengths And Weaknesses:**

This work analyzes several sign-based quantization algorithms for SGD in FL. Sign-based quantization is a well-known approach to compression of model updates and many variants of the technique have been studied. This work presents and analyzes a variant of the technique that outputs the sign after adding noise from some distribution. While, under some assumptions, the technique has advantages over some of the existing techniques the overall novelty of this work is insufficient to recommend acceptance. In addition, a number of relevant works were overlooked before submission suggesting that the paper needs to be reevaluated.